# DNA methylation directs genomic localization of Mbd2 and Mbd3 in embryonic stem cells

Sarah J Hainer[1], Kurtis N McCannell[1], Jun Yu[1], Ly-Sha Ee[1], Lihua J Zhu[1,2,3], Oliver J Rando[4], Thomas G Fazzio[1,3]*

[1]Department of Molecular, Cell, and Cancer Biology, University of Massachusetts Medical School, Worcester, United States; [2]Program in Bioinformatics and Integrative Biology, University of Massachusetts Medical School, Worcester, United States; [3]Program in Molecular Medicine, University of Massachusetts Medical School, Worcester, United States; [4]Department of Biochemistry and Molecular Pharmacology, University of Massachusetts Medical School, Worcester, United States

**Abstract** Cytosine methylation is an epigenetic and regulatory mark that functions in part through recruitment of chromatin remodeling complexes containing methyl-CpG binding domain (MBD) proteins. Two MBD proteins, Mbd2 and Mbd3, were previously shown to bind methylated or hydroxymethylated DNA, respectively; however, both of these findings have been disputed. Here, we investigated this controversy using experimental approaches and re-analysis of published data and find no evidence for methylation-independent functions of Mbd2 or Mbd3. We show that chromatin localization of Mbd2 and Mbd3 is highly overlapping and, unexpectedly, we find Mbd2 and Mbd3 are interdependent for chromatin association. Further investigation reveals that both proteins are required for normal levels of cytosine methylation and hydroxymethylation in murine embryonic stem cells. Furthermore, Mbd2 and Mbd3 regulate overlapping sets of genes that are also regulated by DNA methylation/hydroxymethylation factors. These findings reveal an interdependent regulatory mechanism mediated by the DNA methylation machinery and its readers.

*For correspondence: thomas.fazzio@umassmed.edu

**Competing interests:** The authors declare that no competing interests exist.

## Introduction

In mammalian cells, 5-methylcytosine (5mC), a heritable epigenetic modification on DNA, occurs mainly at CpG dinucleotides through the actions of DNA methyltransferase (DNMT) enzymes (*Bester, 1988*). The members of the DNMT family that modify cytosine on DNA include Dnmt1, Dnmt3a, and Dnmt3b (*Okano et al., 1998*). Dnmt3a and Dnmt3b are de novo DNA methyltransferases, which establish methylation during development. On the other hand, the more abundant Dnmt1 predominantly methylates hemimethylated CpG dinucleotides to restore methylation patterns following genomic replication (although it also has some de novo methylation activity) and therefore, is considered to be the 'maintenance methyltransferase' (*Egger et al., 2006*; *Kho et al., 1998*; *Pradhan et al., 1999*; *Probst et al., 2009*). Dnmt1 is responsible for the majority of methylation in murine embryonic stem (ES) cells, as *Dnmt1* knockout (KO) ES cells carry only about 20% of normal methylation levels (*Lei et al., 1996*).

Active demethylation of 5mC involves a relatively complex series of reactions that starts with oxidation by the ten-eleven translocation (TET) family of dioxygenases (including Tet1, Tet2, and Tet3; (*Lu et al., 2015*), which actively demethylate DNA by oxidizing the 5-methyl group of 5mC to form

5-hydroxymethylcytosine (5hmC) (*Tahiliani et al., 2009*). Further oxidation can occur through conversion of 5hmC into 5-formylcytosine (5fC) and 5-carboxylcytosine (5caC) (*He et al., 2011*; *Ito et al., 2011*). The level of 5hmC is approximately 10% the level of 5mC in ES cells (*Tahiliani et al., 2009*), whereas 5fC and 5caC are much less abundant (*Ito et al., 2011*). Together, Tet1 and Tet2 are responsible for essentially all the 5hmC present in ES cells (*Koh et al., 2011*). However, Tet1 is responsible for 5hmC production at promoter-proximal regions, whereas Tet2 is poorly chromatin-associated and primarily acts within gene bodies (*Huang et al., 2014*; *Vella et al., 2013*). In addition, knockdown (KD) or KO of *Tet1* or *Tet2* skews the profile of ES cell differentiation (*Dawlaty et al., 2011*; *Ficz et al., 2011*; *Koh et al., 2011*), and some reports suggest *Tet1* KD also leads to a defect in self-renewal (*Freudenberg et al., 2012*; *Ito et al., 2010*). However, the functions of Tet proteins during development remain incompletely resolved as *Tet1*[-/-] *Tet2*[-/-] double KO mice have been shown to be viable with only modest phenotypes (*Dawlaty et al., 2013*).

The methylation status of DNA influences many biological processes during mammalian development, and cytosine methylation of promoter-proximal regulatory sequences in mammals is generally considered repressive for gene expression (*Klose and Bird, 2006*). One of the best-studied families of cytosine methylation effectors consists of the methyl-CpG-binding domain (MBD) family of proteins, most of which specifically bind methylated CpG dinucleotides and play important roles in determining the transcriptional state of the mammalian epigenome (*Filion et al., 2006*; *Hendrich and Bird, 1998*; *Prokhortchouk et al., 2001*; *Unoki et al., 2004*). MBD proteins are transcriptional regulators, primarily thought to act as repressors, which play a major role in coordinating crosstalk between cytosine methylation and chromatin structure to regulate transcription (*Denslow and Wade, 2007*). The MBD has the ability to bind single symmetrically methylated CpG dinucleotides (*Nan et al., 1993*; *Ohki et al., 2001*), and the functions of MeCP2 and the Mbd1-4 family members have been well-studied. With the exception of Mbd3, these proteins preferentially bind 5mC over unmethylated cytosine (*Hendrich and Bird, 1998*; *Zhang et al., 1999*). Consistent with this binding activity, Mbd2 occupies methylated CpG island–containing promoters of inactive genes (*Chatagnon et al., 2011*; *Menafra et al., 2014*). Although Mbd3 does not bind 5mC, MeCP2 and Mbd3 can bind 5hmC in vitro and Mbd3 is enriched in regions of elevated 5hmC in ES cells (*Mellén et al., 2012*; *Yildirim et al., 2011*). Consistent with this reported localization, knockdown (KD) of *Mbd3* leads to misregulation of 5hmC-marked genes (*Yildirim et al., 2011*). Furthermore, Mbd3 binding in ESCs was strongly reduced upon RNAi-mediated KD of *Tet1* (*Yildirim et al., 2011*) However, arguing against the above studies, other in vitro studies using short DNA probes containing a single symmetric 5hmCpG find poor binding of 5hmC by MBD family members (*Cramer et al., 2014*; *Spruijt et al., 2013*).

Mbd2 and Mbd3 are highly similar in amino acid sequence and are components of mutually exclusive versions of the nucleosome remodeling and deacetylase (NuRD) complex (*Hendrich and Bird, 1998*; *Le Guezennec et al., 2006*; *Wade et al., 1999*; *Zhang et al., 1999*). Furthermore, Mbd2 and Mbd3 exhibit partially overlapping localization profiles at some methylated regions in vivo (*Günther et al., 2013*), consistent with the possibility that these factors bind to DNA enriched for 5mC or its derivative, 5hmC. However, these highly similar complexes play distinct biological roles in vivo. Mbd3/NuRD is necessary for ES cell pluripotency and differentiation, as well as embryonic development, whereas *Mbd2* KO mice are viable and fertile (*Hendrich et al., 2001*; *Kaji et al., 2006*; *Reynolds et al., 2012*). In addition, Mbd3/NuRD coordinates cytosine methylation by recruiting DNA methyltransferases to the promoters of tumor suppressor genes in colon cancer cell and leukemia cell lines (*Cai et al., 2014*; *Choi et al., 2013*; *Morey et al., 2008*), and depletion of *Mbd3* results in reduced DNA methylation levels at some locations in ES cells (*Latos et al., 2012*).

Recently, evidence has arisen questioning the dependence of Mbd2 and Mbd3 on cytosine methylation for genomic localization (*Baubec et al., 2013*). The authors of this study reported that the enrichments of Mbd2 and Mbd3 at LMRs (low-methylated regions that are enriched for transcription factor-binding sites and exhibit approximately 30% methylation on average) (*Stadler et al., 2011*) were minimally altered in *Dnmt1*[-/-] *Dnmt3a*[-/-] *Dnmt3b*[-/-] triple knockout (TKO) ES cells. Here, we sought to resolve the conflicting data addressing the dependence of Mbd2 and Mbd3 localization on 5mC and 5hmC. Analyses of ChIP-seq data from Baubec et al. as well as multiple new ChIP-seq datasets reported here demonstrate methylation-dependence of Mbd2 and Mbd3 binding throughout the genome. Interestingly, we show that Mbd2 and Mbd3 exhibit significantly overlapping localization in vivo and find that Tet1 activity is required for normal chromatin association by both Mbd3

and Mbd2. Furthermore, we show that Mbd3 and Mbd2 are each required for the binding of the other, as well as for normal levels of 5mC and 5hmC. Finally, we find that individual KD of *Mbd2*, *Mbd3, Dnmt1,* or *Tet1* results in highly concordant changes in gene expression. Together these data reveal interdependence among Mbd2/NuRD, Mbd3/NuRD, 5mC, and 5hmC to maintain normal chromatin structure and gene regulation in ESCs.

## Results

### Evidence of methylation-dependent localization of Mbd3 and Mbd2 in bio-MBD datasets

Multiple studies have established that Mbd2, but not Mbd3, binds methylated DNA with higher affinity than unmethylated DNA (*Hendrich and Bird, 1998*; *Zhang et al., 1999*). We previously found that Mbd3 occupies genomic regions containing 5hmC in ES cells (*Yildirim et al., 2011*). Furthermore, we found that KD of *Tet1* results in a reduction of Mbd3 occupancy, and that KD of *Mbd3* results in a reduction in bulk 5hmC levels (*Yildirim et al., 2011*), demonstrating interdependence between Mbd3 and 5hmC. These findings also lead to a straightforward prediction that cytosine methylation would be required for Mbd3 binding, since 5hmC is generated by oxidation of 5mC. However, Baubec et al. reported that both Mbd3 and Mbd2 occupancies at LMRs are minimally altered in TKO cells (*Baubec et al., 2013*), arguing against our previous findings regarding Mbd3 and long-standing models of Mbd2 function.

To assess the discrepancies between these studies, we first re-analyzed the datasets from Baubec et al. (*Figure 1*, *Figure 1—figure supplement 1*, and *Figure 1—figure supplement 2*). There were two ChIP-seq replicates for the biotin-tagged *Mbd3a* isoform (bio-Mbd3a) in wild-type (WT) cells and a single replicate reported for bio-Mbd3a ChIP-seq in TKO cells (see *Table 1*). We initially focused on genomic binding at the subset of LMRs that are larger than 150 bp and at least 3 kb away from another LMR or an unmethylated region (hereafter denoted as 'LMR subset') that was utilized in the prior study. We observed only a modest reduction (~28%) in bio-Mbd3a occupancy at the LMR subset upon loss of DNA methylation in TKO cells (*Figure 1A*), similar to the original observation (see Figure 7A of *Baubec et al. (2013)*). However, when we assessed the change in overall bio-Mbd3a enrichment by examining all bio-Mbd3a peaks (called from pooled WT bio-Mbd3a ChIP-seq reads), we observed a ~75% reduction in bio-Mbd3a binding in TKO cells relative to WT cells (*Figure 1B–C* and *Figure 1—figure supplement 1A–B*). These data demonstrate that Mbd3a relies heavily on DNA methylation in order to bind chromatin in ES cells and that confining analysis to a subset of LMRs obscures the global impact of methylation on Mbd3 localization.

While Baubec et al. found that bio-Mbd2 requires DNA methylation for proper binding to chromatin at highly methylated CpG islands (Figure 4G–H of *Baubec et al. (2013)*), they observed minimal effect of loss of methylation on bio-Mbd2 binding at the LMR subset (Figure 7A from *Baubec et al. (2013)*). To replicate this prior analysis, we pooled the mapped reads for each of three bio-Mbd2 ChIP-seq replicates from WT or two bio-Mbd2 ChIP-seq replicates from TKO cells (see *Table 1*) and measured their enrichment over the LMR subset. We observed a modest reduction (~15%) in bio-Mbd2 occupancy relative to WT (*Figure 1D*), consistent with the prior report. In contrast, when we analyzed bio-Mbd2 binding over all LMRs (referred to throughout as 'total LMRs'), we observed a stronger reduction (~40%) in bio-Mbd2 occupancy in the pooled TKO libraries relative to WT (*Figure 1E*).

To better evaluate the effects of cytosine methylation on bio-Mbd2 binding, we analyzed each bio-Mbd2 ChIP-seq replicate separately. We observed high variation among the three replicates for bio-Mbd2 in WT cells (*Figure 1F–H* and *Figure 1—figure supplement 1C–I*). For example, while two WT replicates are modestly enriched for bio-Mbd2 binding relative to the two TKO replicates (~10% reduction) at the LMR subset, the third bio-Mbd2 ChIP-seq experiment in WT cells has a much higher peak, implying a much larger (~50%) reduction in occupancy in TKO cells (*Figure 1F*). Similar disparities were observed at other regions of the genome (*Figure 1—figure supplement 1C–F*). Importantly, when we examined the effect of 5mC loss over all peaks of bio-Mbd2 binding, bio-Mbd2 enrichment was reduced to near background levels in TKO cells (*Figure 1G*). Furthermore, we observed considerable gene-by-gene differences in the binding profiles among WT bio-Mbd2

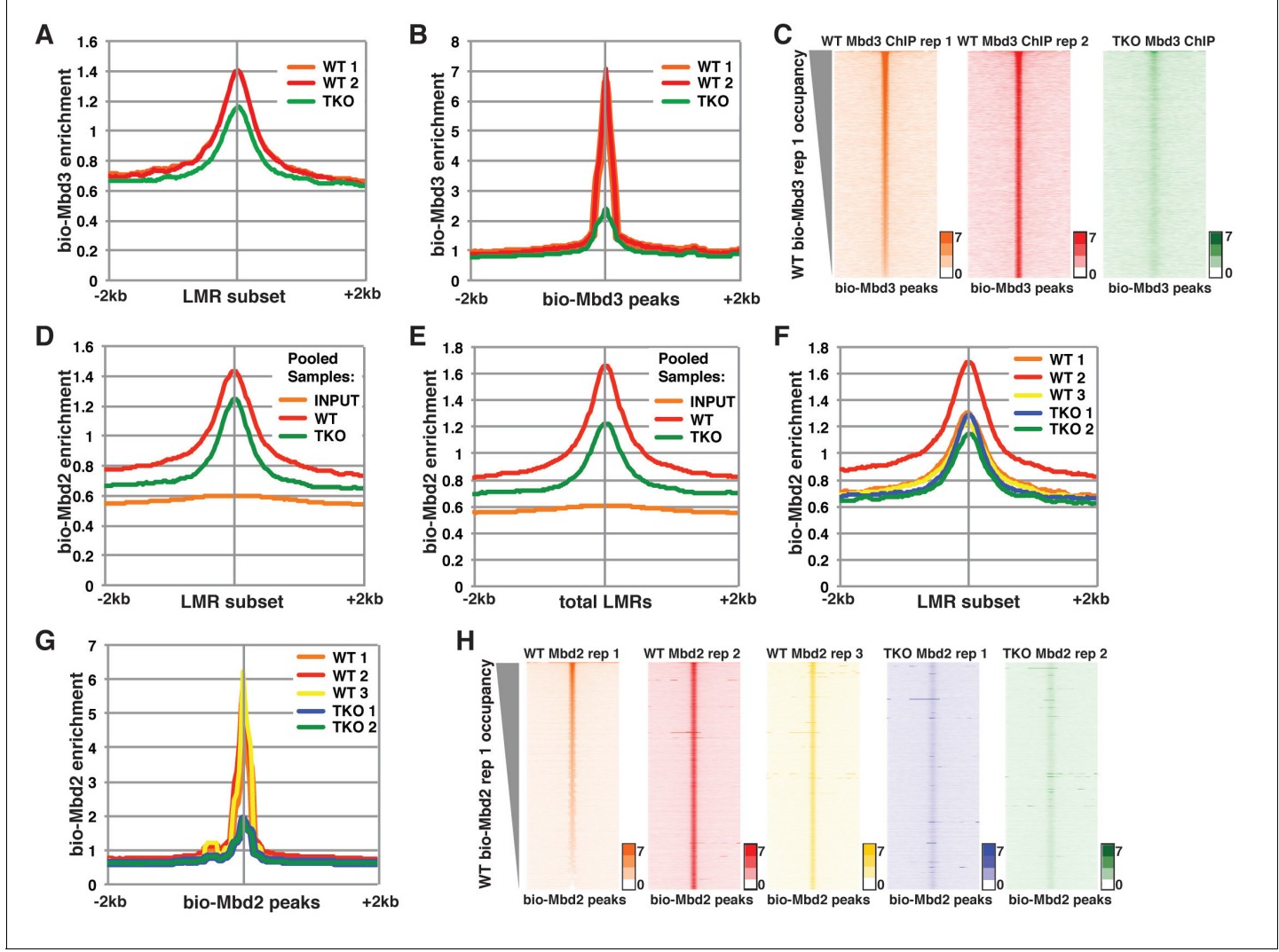

**Figure 1.** Analysis of bio-Mbd3 and bio-Mbd2 ChIP-seq datasets reveals methylation-dependent chromatin localization in ES cells. Mbd3 and Mbd2 ChIP-seq datasets from (*Baubec et al., 2013*) were obtained from GEO (GSE39610; see *Table 1*). (A–C) Analysis of ChIP-seq data for biotin-tagged Mbd3 (bio-Mbd3) in WT or TKO cells. The two available WT replicates and one available TKO replicate were mapped over the LMR subset (adapted from *Stadler et al. [2011]*) ± 2 kb (A) or bio-Mbd3 peaks from WT cells ± 2 kb (B). (C) Heatmaps for bio-Mbd3 ChIP-seq in WT or TKO cells over bio-Mbd3 peaks ± 2 kb, sorted by bio-Mbd3 occupancy (high to low) in replicate 1 WT cells. (D–H) Analysis of ChIP-seq data for biotin-tagged Mbd2 (bio-Mbd2) in WT or TKO cells. (D–E) Data were analyzed as described in (*Baubec et al., 2013*) where replicates were pooled and mapped over LMRs that are > 150 bp and at least 3 kb away from other LMRs or unmethylated regions (LMR subset) (D) or all LMRs (E). (F–G) Biological replicates were analyzed separately and mapped over the LMR subset ± 2 kb (F) or bio-Mbd2 peaks from WT cells ± 2 kb (G). (H) Heatmaps for bio-Mbd2 ChIP-seq in WT or TKO cells over bio-Mbd2 peaks ± 2 kb, sorted by bio-Mbd2 occupancy in replicate 1 WT cells. Peaks of bio-Mbd3 and bio-Mbd2 were called as described in Supplemental Experimental Procedures.

The following figure supplements are available for figure 1:

**Figure supplement 1.** Variable bio-Mbd2 binding in WT cells over TSSs and gene-distal DHSs.

**Figure supplement 2.** Variable bio-Mbd1 and bio-MeCP2 binding in WT cells.

replicates, where individual locations that are lowly bound in one replicate are highly bound in another replicate (*Figure 1H*, *Figure 1—figure supplement 1G–I*). Taken together, these analyses demonstrate bio-Mbd3 and bio-Mbd2 strongly depend on cytosine methylation for normal chromatin association. Furthermore, poor concordance between biological replicates was also found for several

**Table 1.** Related to *Figure 1*. SRA file numbers for re-analyzed *Baubec et al. (2013)* datasets.

| Name | SRA file number | Number of mapped reads (Bowtie, up to three mismatches) |
| --- | --- | --- |
| WT Mbd3 ChIP replicate 1 | SRR696667 | 11,601,021 |
| WT Mbd3 ChIP replicate 2 | SRR696673 | 14,166,151 |
| TKO Mbd3 ChIP | SRR769560 | 34,666,789 |
| WT Mbd2 ChIP replicate 1 | SRR527128 | 9,441,721 |
| WT Mbd2 ChIP replicate 2 | SRR527129 | 15,116,380 |
| WT Mbd2 ChIP replicate 3 | SRR696658 | 12,545,659 |
| TKO Mbd2 ChIP replicate 1 | SRR527161 | 24,282,816 |
| TKO Mbd2 ChIP replicate 2 | SRR696682 | 7,605,171 |
| WT Mbd1a ChIP replicate 1 | SRR527126 | 11,566,092 |
| WT Mbd1a ChIP replicate 2 | SRR527127 | 19,528,778 |
| TKO Mbd1a ChIP | SRR527159 | 22,070,693 |
| WT Mbd1b ChIP replicate 1 | SRR527147 | 14,320,842 |
| WT Mbd1b ChIP replicate 2 | SRR527148 | 22,371,797 |
| TKO Mbd1b ChIP | SRR527160 | 7,503,068 |
| WT Mbd4 ChIP replicate 1 | SRR527131 | 13,864,108 |
| WT Mbd4 ChIP replicate 2 | SRR527132 | 23,655,753 |
| TKO Mbd4 ChIP replicate 1 | SRR669321 | 13,670,155 |
| TKO Mbd4 ChIP replicate 2 | SRR669322 | 11,251,853 |
| WT MeCP2 ChIP replicate 1 | SRR527133 | 19,229,199 |
| WT MeCP2 ChIP replicate 2 | SRR527134 | 11,771,499 |
| WT MeCP2 ChIP replicate 3 | SRR696681 | 12,118,572 |
| TKO MeCP2 ChIP replicate 1 | SRR527162 | 17,003,233 |
| TKO MeCP2 ChIP replicate 2 | SRR696683 | 28,586,006 |

other MBD proteins profiled in Baubec et al. (*Figure 1—figure supplement 2*), preventing straightforward interpretation of the role of cytosine methylation in localization of these factors.

## *Dnmt1* and *Tet1* are required for Mbd3 and Mbd2 occupancy in ES cells

To further address the discrepancies discussed above, and to better elucidate the roles of DNA methylation and hydroxymethylation in chromatin binding by Mbd3 and Mbd2, we performed biological duplicate ChIP-seq experiments for endogenous Mbd3 and Mbd2 in ES cells acutely depleted of *Dnmt1* or *Tet1* (*Figure 2*). Upon efficient KD of these factors (*Figure 2—figure supplement 1A–B*), we observed a significant reduction in promoter-proximal Mbd3 and Mbd2 binding relative to the binding observed in control (*EGFP* KD) cells (examples shown in *Figure 2A*, *Figure 2—figure supplement 1C–D*). The reductions in Mbd3 and Mbd2 binding observed upon *Dnmt1* KD or *Tet1* KD were not due to altered expression of these proteins (*Figure 2—figure supplement 1E*). When we compared the locations bound by Mbd3 and Mbd2 throughout the genome, we found that Mbd3 and Mbd2 co-occupy numerous regions throughout the genome of ES cells, and this overlap is particularly enriched at promoter-proximal regions (*Figure 2B*).

Next, we quantified average binding profiles of Mbd3 and Mbd2 over six genomic features – intermediate CpG density promoters (ICPs; from (*Weber et al., 2007*), transcription start sites (TSSs), the LMR subset described above, gene-distal DNaseI hypersensitive sites (DHSs; from ENCODE; GSM1014154), Mbd2 peaks (from *EGFP* KD cells), and Mbd3 peaks (from *EGFP* KD cells). We observed a similar pattern over each genomic feature: *Dnmt1* KD and *Tet1* KD each resulted in a dramatic reduction in Mbd3 and Mbd2 occupancies (*Figure 2C–F*). Importantly, our duplicate ChIP-seq datasets were highly reproducible (*Figure 2A*; *Figure 2—figure supplement 1C–D* and *Figure 2—figure supplement 2A–D*). We validated our observations at eight genomic locations by

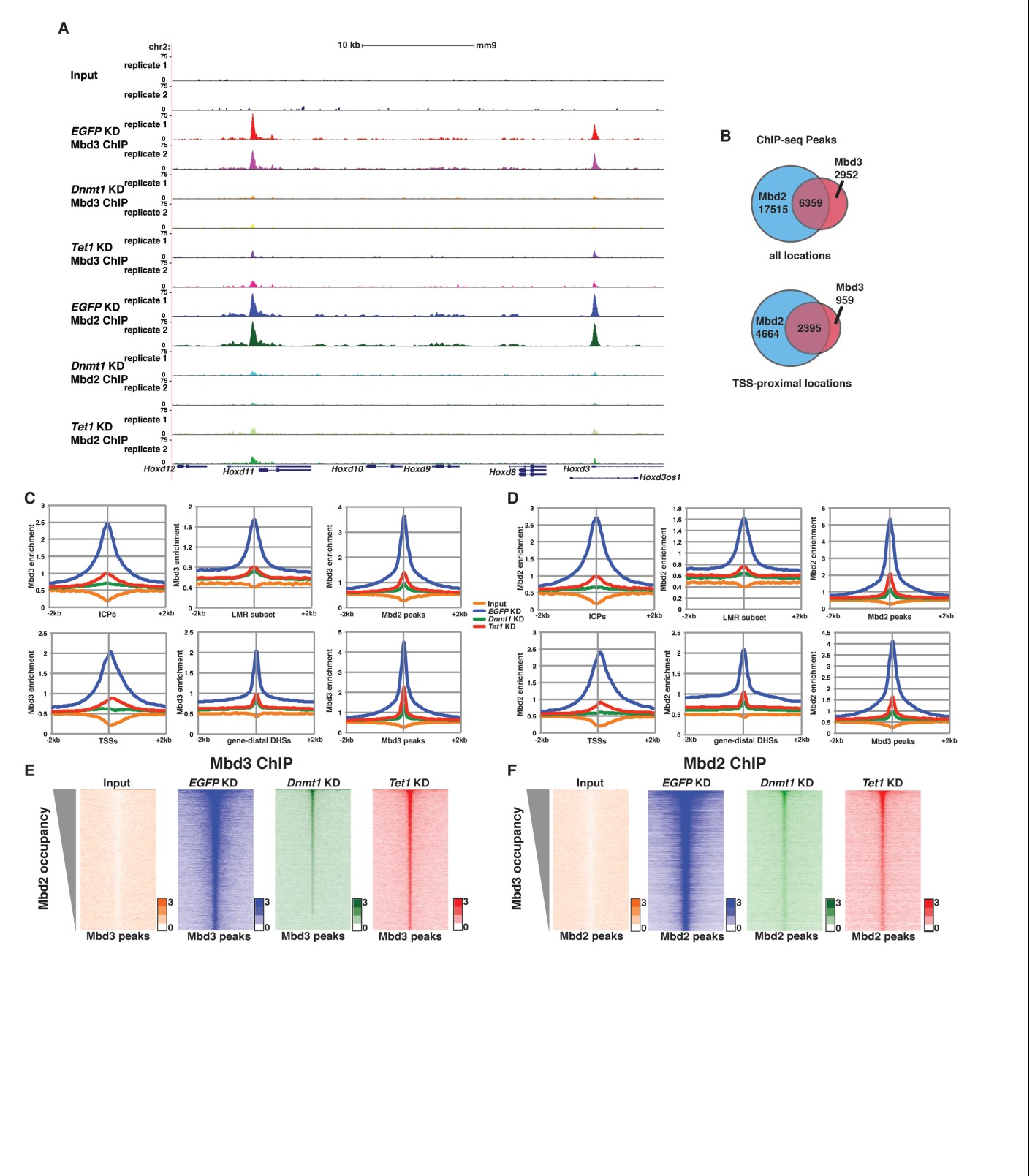

**Figure 2.** *Dnmt1* and *Tet1* are required for Mbd3 and Mbd2 binding in ES cells. (**A**) Genome browser tracks of replicate ChIP-seq experiments examining endogenous Mbd3 or Mbd2 occupancy in control (*EGFP* KD), *Dnmt1* KD, and *Tet1* KD ES cells over indicated loci (*Hoxd* cluster). (**B**) Overlap of Mbd3 and Mbd2 binding. Shown are Venn diagrams delineating the overlap between all genomic locations (top panel) or TSS-proximal locations (−1 kb to +100 bp; bottom panel) bound by Mbd3 and Mbd2. (**C–D**) Aggregation plots of Mbd3 (**C**) or Mbd2 (**D**) ChIP-seq data showing occupancy

*Figure 2 continued on next page*

*Figure 2 continued*

over ICPs (top left panel, from [**Weber et al., 2007**]), annotated TSSs (bottom left panel), the LMR subset (top middle panel, from [**Stadler et al., 2011**]), gene-distal DHSs (bottom middle panel, from GSM1014154 with TSSs removed) Mbd2 peaks (top right panel, called from *EGFP* KD ChIP-seq experiments), and Mbd3 peaks (bottom right panel, called from *EGFP* KD ChIP-seq experiments) ± 2 kb in control (*EGFP* KD), *Dnmt1* KD, or *Tet1* KD ES cells. (E–F) Heatmaps of Mbd3 enrichment over Mbd3 binding sites sorted by Mbd2 occupancy (E) and Mbd2 enrichment over Mbd2 binding sites sorted by Mbd3 occupancy (F) in control (*EGFP* KD), *Dnmt1* KD, or *Tet1* KD ES cells. The profiles shown in aggregation plots and heatmaps represent the average of two biological replicates.

The following figure supplements are available for figure 2:

**Figure supplement 1.** Loss of *Dnmt1* and *Tet1* results in reduced occupancy of both Mbd3 and Mbd2 in ES cells.

**Figure supplement 2.** Validation of endogenous Mbd3 and Mbd2 ChIP-seq experiments.

**Figure supplement 3.** FLAG ChIPs confirm Dnmt1 and Tet1 are required for Mbd3 and Mbd2 occupancies.

**Figure supplement 4.** ChIP-seq experiments using MBD-FLAG fusions.

---

performing three biological replicate ChIP-qPCRs for Mbd3 and Mbd2 in *Dnmt1* KD and *Tet1* KD ES cells (*Figure 2—figure supplement 2E*) and in *Dnmt1* KO cells (*Figure 2—figure supplement 2F–G*).

Four key controls demonstrate the specificity of these ChIP data. First, we performed ChIP-qPCR for Mbd3 in *Mbd3* KD cells and for Mbd2 in *Mbd2* KD cells, confirming the reduced ChIP signals expected in both cases (*Figure 2—figure supplement 2E*). To validate our findings in a manner independent of the antibodies targeting the endogenous proteins, we genetically FLAG-tagged endogenous Mbd2 (hereafter *Mbd2-3XFLAG*) or Mbd3 ([*Yildirim et al., 2011*]; hereafter *Mbd3abc-3XFLAG*) and performed ChIP-qPCR or ChIP-seq, respectively, in *EGFP* KD, *Dnmt1* KD, and *Tet1* KD cells (*Figure 2—figure supplement 3A–D*). We again observed a reduction in Mbd2-3XFLAG and Mbd3abc-3XFLAG enrichment following knockdown of *Dnmt1* or *Tet1* (*Figure 2—figure supplement 3B–D*), despite lower overall enrichment than we observed with antibodies against the endogenous proteins. FLAG ChIP-seq and ChIP-qPCR for Mbd3abc-3XFLAG in *Mbd3* KD cells and for Mbd2-3XFLAG in *Mbd2* KD cells confirmed the reduced ChIP signals expected in both cases (*Figure 2—figure supplement 3C–D*; *Figure 2—figure supplement 4A–B*). Finally, to test whether overexpression of *Mbd3a* (analogous to the bio-Mbd3a ChIP-seq studies in [*Baubec et al., 2013*]) resulted in altered genomic binding relative to endogenously expressed *Mbd3*, and whether chromatin binding by overexpressed Mbd3a was altered upon *Dnmt1* KD or *Tet1* KD, we overexpressed *Mbd3a-3XFLAG* in ES cells and performed ChIP-qPCR experiments upon KD of *Dnmt1*, *Tet1*, or *EGFP* (*Figure 2—figure supplement 3C*). Although Mbd3a-3XFLAG enrichment was reduced relative to endogenously expressed Mbd3abc-3XFLAG, there was a decrease in Mbd3a-3XFLAG binding upon *Dnmt1* or *Tet1* depletion at all eight promoter-proximal regions examined (*Figure 2—figure supplement 3C*). Together, these data demonstrate a requirement for *Dnmt1* and *Tet1* for Mbd3 and Mbd2 binding throughout the genome of ES cells. Although the dependence of Mbd3 and Mbd2 on DNA methylation (or *Dnmt1*) and the requirement of Mbd3 for hydroxymethylation (or *Tet1*) were consistent with our previous findings and those of others (*Chatagnon et al., 2011*; *Menafra et al., 2014*; *Yildirim et al., 2011*), the requirement of Mbd2 for *Tet1* has not been previously observed. We will address this finding below.

One argument suggesting the data in Yildirim et al. may not be valid was that Mbd3 binding was not observed at enhancers (*Menafra and Stunnenberg, 2014*). In this study, we used H3K4me1 as a marker for enhancer regions (*Yildirim et al., 2011*), which is an imperfect marker for putative enhancer elements (*Shlyueva et al., 2014*). We therefore re-analyzed the Yildirim data at gene-distal DNaseI hypersensitive sites (DHSs), which represent active enhancers and some additional regulatory features, and observed a peak of Mbd3 binding that was strongly reduced upon *Tet1* depletion (*Figure 2—figure supplement 4C*). Similarly, our new Mbd3 and Mbd2 ChIP-seq datasets exhibit robust enrichment at gene-distal DHSs that is reduced upon *Dnmt1* KD or *Tet1* KD (*Figure 2C–D*),

confirming Mbd3 binds to these regions in a manner dependent on the cytosine methylation and hydroxymethylation machinery.

## The catalytic activity of Tet1 is required for proper chromatin localization by Mbd3 and Mbd2 in ES cells

Although our KD studies demonstrated *Tet1* is necessary for chromatin binding by Mbd3 and Mbd2, the Tet1 protein has been shown, in some cases, to regulate gene expression independently of its catalytic activity (*Jin et al., 2014*; *Kaas et al., 2013*; *Williams et al., 2011*). These findings leave open the possibility that Tet1 functions in Mbd3 and Mbd2 binding independently of its roles in DNA hydroxymethylation and demethylation. To test whether the catalytic activity of Tet1 is necessary for Mbd3 and Mbd2 binding in ES cells, we generated homozygous catalytically inactive *Tet1* mutant (*Tet1c.i.*) ES cells. We knocked-in two amino acid substitutions, H1652Y and D1654A (*Figure 3—figure supplement 1*), which were previously demonstrated to eliminate DNA demethylation activity by Tet1 (*Ito et al., 2010*; *Tahiliani et al., 2009*). We performed ChIP-seq for Mbd3 and Mbd2 in two independent *Tet1c.i.* lines (*Figure 3*). We found that relative to WT ES cells, ChIP-seq of Mbd3 and Mbd2 in *Tet1c.i.* cells shows decreased occupancies of both Mbd3 and Mbd2 at ICPs, TSSs, the LMR subset, gene-distal DHSs, Mbd2 peaks, and Mbd3 peaks (*Figure 3*). These data demonstrate that the catalytic activity of Tet1 is required to promote binding of Mbd3 and Mbd2 to multiple genomic locations in ES cells.

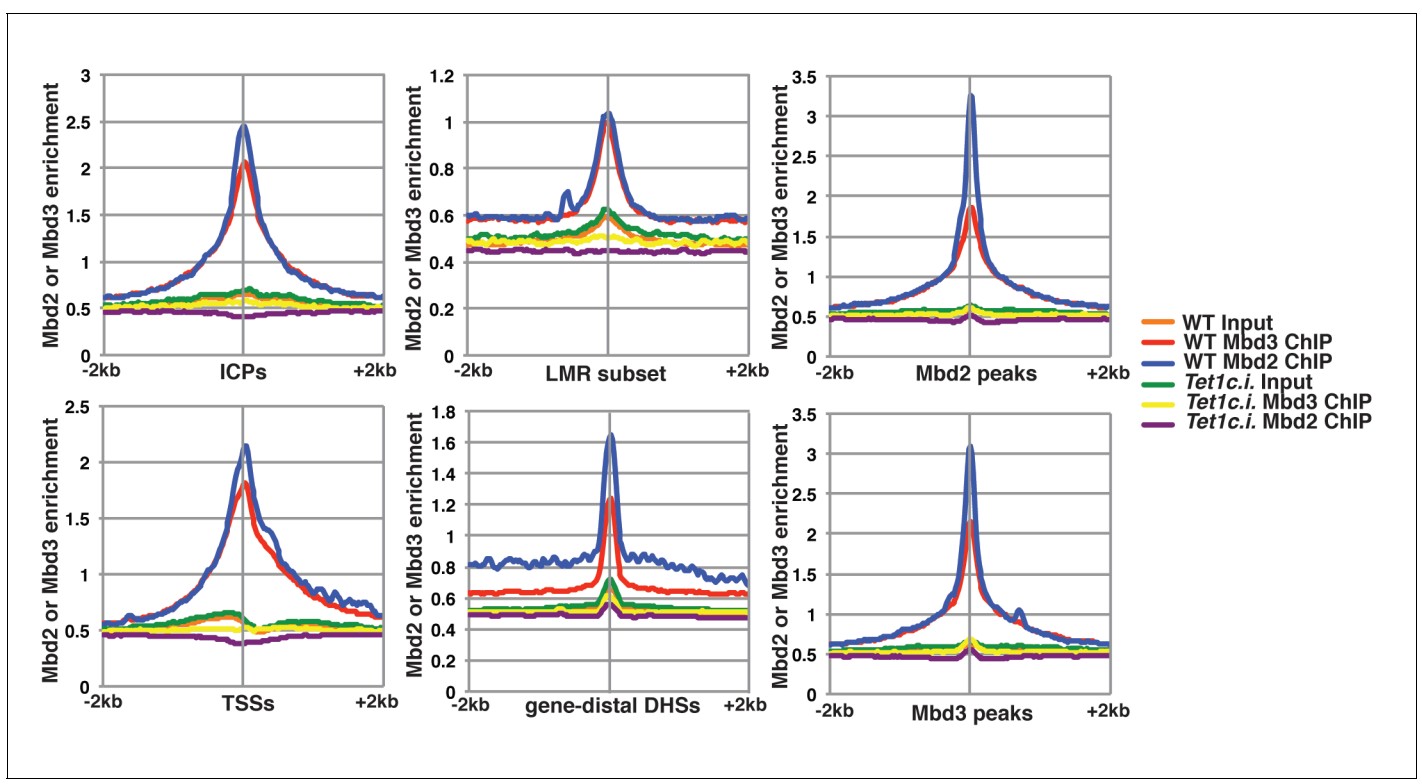

**Figure 3.** The catalytic activity of Tet1 is required for occupancy of Mbd3 and Mbd2 in ES cells. Aggregation plots of Mbd3 or Mbd2 ChIP-seq showing association over ICPs (top left panel), annotated TSSs (bottom left panel), the LMR subset (top middle panel), gene-distal DHSs (bottom middle panel), Mbd2 peaks (top right panel), and Mbd3 peaks (bottom right panel) ± 2 kb in wild-type (WT) and *Tet1* catalytically inactive (*Tet1c.i.*) ES cells. Aggregation plots represent the average of two biological replicates.

The following figure supplement is available for figure 3:

**Figure supplement 1.** Catalytic mutation in *Tet1* does not alter total protein levels.

## Loss of *Mbd3* or *Mbd2* results in reduced 5mC and 5hmC at regulatory regions

In order to better understand the dependence of Mbd3 and Mbd2 on cytosine methylation and hydroxymethylation, we performed duplicate MeDIP-seq and hMeDIP-seq experiments (which enrich for DNA harboring 5mC or 5hmC, respectively) on *EGFP* KD, *Dnmt1* KD, *Mbd3* KD, and *Mbd2* KD ES cells (*Figure 4A–B*). We examined the effects of depletion of these factors on methylation (*Figure 4A*) and hydroxymethylation (*Figure 4B*) at ICPs, TSSs, the LMR subset, gene-distal DHSs, Mbd2 peaks, and Mbd3 peaks. As expected, *Dnmt1* KD resulted in decreased 5mC and 5hmC. Furthermore, both *Mbd3* KD and *Mbd2* KD also resulted in decreased 5mC and 5hmC levels. These data are consistent with previous findings showing that *Mbd3* KO cells have decreased 5mC levels at some locations (*Latos et al., 2012*), and reduced 5hmC levels overall (*Yildirim et al., 2011*) in ES cells. Interestingly, when we examined the methylation and hydroxymethylation profiles over total LMRs (*Figure 4—figure supplement 1A–B*), we observed a much stronger peak of enrichment in control (*EGFP* KD) ES cells than found at the LMR subset. Further investigation revealed that the LMRs excluded from the LMR subset were overrepresented at promoter proximal regions of genes (*Figure 4—figure supplement 1E*), and exhibited higher average methylation and hydroxymethylation (*Figure 4—figure supplement 1C–D*).

To validate these data, we performed dot blotting, thin layer chromatography (TLC), and digests with methylation sensitive (HpaII) or insensitive (MspI) restriction enzymes on DNA isolated from *Mbd3* KD, *Mbd2* KD, or control cells (*EGFP* KD and *Dnmt1* KD). Consistent with the MeDIP- and hMeDIP-seq data, bulk 5mC and 5hmC levels were reduced upon *Mbd3* KD or *Mbd2* KD, as measured by dot blotting and TLC, while digestion with HpaII was enhanced (*Figure 4C–F*). Together these data demonstrate that levels of cytosine methylation and hydroxymethylation are regulated not only by the DNA methyltransferase enzymes, but also by factors that depend on these modifications for proper chromatin localization.

## Interdependence of Mbd3 and Mbd2 occupancy

The reduction in 5mC and 5hmC levels observed upon depletion of *Mbd3* and *Mbd2* raises the possibility that Mbd3 and Mbd2 may each be necessary (either directly or indirectly) for normal chromatin binding by the other. To address this possibility, we investigated the effect of *Mbd3* KD or *Mbd2* KD on chromatin association by Mbd2 or Mbd3, respectively. Relative to control (*EGFP* KD) cells, we found that when *Mbd3* was depleted there was a decrease in Mbd2 occupancy, and, conversely, when Mbd2 was depleted there was a decrease in Mbd3 occupancy (example loci shown in *Figure 5A*; *Figure 5—figure supplement 1A–B*). We extended this analysis to the whole genome, confirming a significant reduction in binding of both Mbd3 and Mbd2 upon KD of the other MBD protein at ICPs, TSSs, the LMR subset, gene-distal DHSs, Mbd2 peaks, and Mbd3 peaks (*Figure 5B–C*). Depletion of *Mbd3* does not alter Mbd2 levels, nor does depletion of *Mbd2* alter levels of Mbd3 (*Figure 2—figure supplement 1E*). We validated our observations by performing three biological replicate ChIP-qPCRs for Mbd3 and Mbd2 in *Mbd3* KD and *Mbd2* KD ES cells (*Figure 2—figure supplement 2E*) and by ChIP-seq for Mbd3abc-3XFLAG and Mbd2-3XFLAG in *Mbd3* KD and *Mbd2* KD ES cells (*Figure 2—figure supplement 4A–B*). Together these data demonstrate interdependence between Mbd3 and Mbd2, whereby these mutually exclusive constituents of the NuRD complex regulate the genomic localization of one another.

Mbd3/NuRD was previously shown to play a role in methylation of promoter-proximal regions in cancer cell lines by helping recruit DNA methyltransferase enzymes to specific loci (*Cai et al., 2014*; *Choi et al., 2013*; *Morey et al., 2008*). To investigate whether this potential mechanism of regulation was also operational in ES cells, we examined the binding profile of Dnmt1 in *Mbd3* KO cells (*Figure 5—figure supplement 2*). We found that *Mbd3* KO cells (*Figure 5—figure supplement 2A*) showed a general loss of Dnmt1 binding, with Dnmt1 occupancy decreased relative to WT cells at all genomic locations examined (*Figure 5—figure supplement 2B*). These data show Dnmt1 relies on *Mbd3* for proper binding. Therefore, the decreased methylation, hydroxymethylation, and Mbd2 binding observed upon depletion of *Mbd3* may be attributed (at least in part) to loss of chromatin binding by Dnmt1.

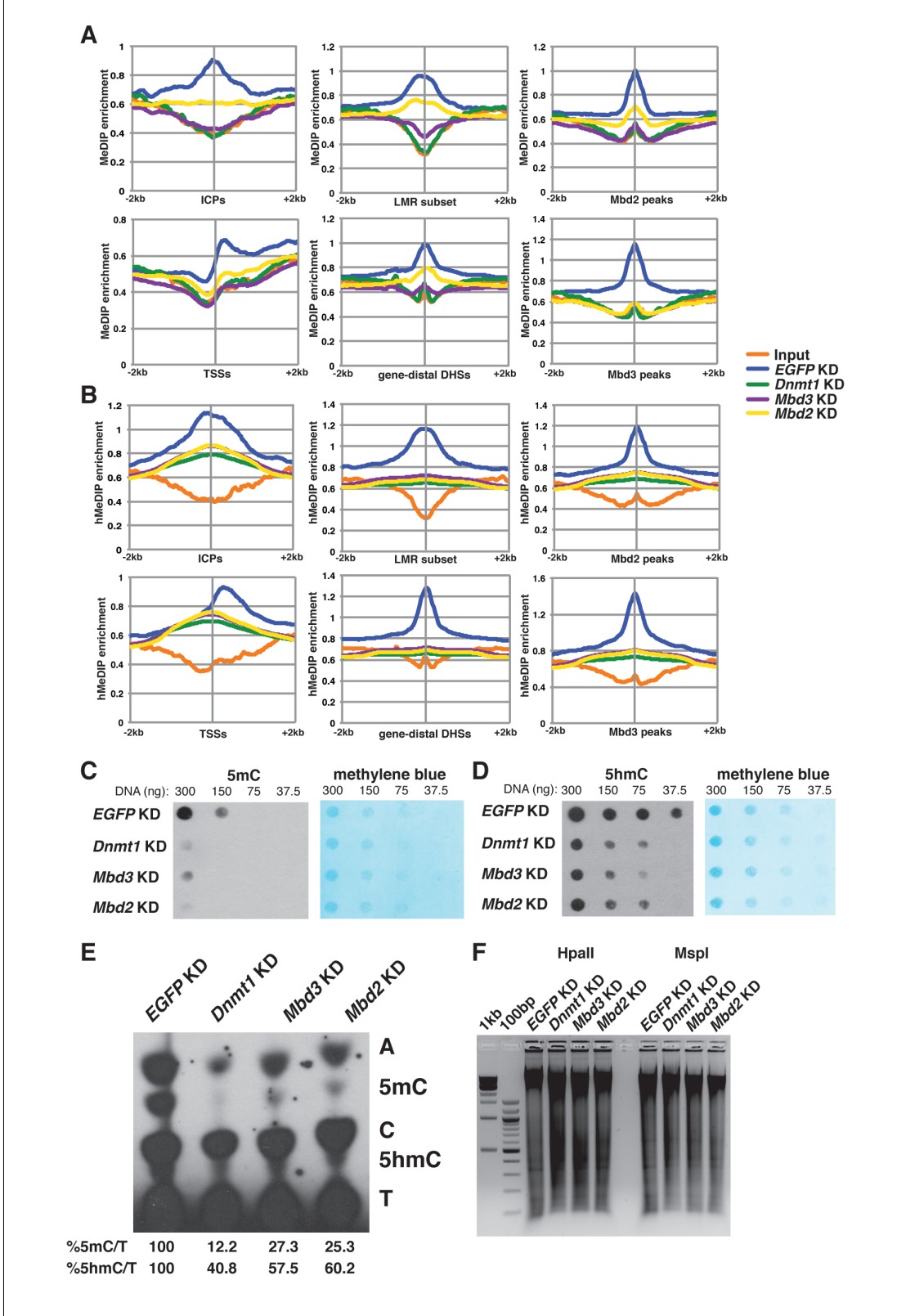

**Figure 4.** Dnmt1, Mbd3, and Mbd2 are required for 5mC and 5hmC in ES cells. (**A–B**) Aggregation plots of 5mC (**A**) or 5hmC (**B**) enrichment over ICPs (top left panel), annotated TSSs (bottom left panel), the LMR subset (top middle panel), gene-distal DHSs (bottom middle panel) Mbd2 peaks (top right panel), and Mbd3 peaks (bottom right panel) ± 2 kb in control (*EGFP* KD), *Dnmt1* KD, *Mbd3* KD, or *Mbd2* KD ES cells. Aggregation plots represent the average of two biological replicates. (**C–D**) Dot blot of bulk 5mC (**C**) and 5hmC (**D**) levels in *EGFP* KD, *Dnmt1* KD, *Mbd3* KD, or *Mbd2* KD ES cells. Left

Figure 4 continued

panel shows 5mC or 5hmC levels and right panel shows methylene blue staining, which serves as a loading control. (E) Thin layer chromatography separation of radioactively end-labeled bases from MspI digested genomic DNA. Quantification of 5mC or 5hmC is expressed relative to T in each KD. Levels in *EGFP* KD are set to 100%. (F) Restriction enzyme digest of genomic DNA from *EGFP* KD, *Dnmt1* KD, *Mbd3* KD, or *Mbd2* KD ES cells using an enzyme blocked by CpG methylation/hydroxymethylation (HpaII) or a CpG methylation insensitive enzyme (MspI) that cuts the same site (CCGG).

The following figure supplement is available for figure 4:

**Figure supplement 1.** 5mC and 5hmC levels over total and removed LMRs.

## Overlapping effects of *Dnmt1*, *Tet1*, *Mbd3*, or *Mbd2* loss on gene expression

The various interdependencies detailed above make an additional functional prediction – that *Dnmt1*, *Tet1*, *Mbd3*, and *Mbd2* should have overlapping roles in gene regulation in ES cells. To test this, we used DNA microarrays to examine the changes in mRNA levels upon *Dnmt1* KD, *Tet1* KD, *Mbd3* KD, *Mbd2* KD, and (since it is a key constituent of both NuRD complexes in ES cells) *Chd4* KD. We observed well-correlated gene expression profiles in ES cells knocked down individually for these factors (*Figure 6—figure supplement 1*, Table 3), suggesting a significant overlap in their sets of target genes, although *Tet1* KD and *Mbd2* KD generally had weaker effects on gene expression (*Figure 6A*), consistent with the less severe phenotypes observed upon *Mbd2* loss (*Günther et al., 2013*; *Hendrich et al., 2001*). Together, these data are consistent with a model in which these chromatin remodeling factors and DNA modifications coordinate to control gene activity in ES cells.

## Discussion

The role of cytosine methylation in the regulation of Mbd3 and Mbd2 binding to chromatin has been controversial. Here, we sought to address these conflicting data. Several differences in study design may help explain some of the discrepancies between our previous studies and those reported by Baubec et al. First, whereas endogenous Mbd3 was examined in ChIP studies performed in Yildirim et al., the bio-tagged Mbd3 in Baubec et al. was overexpressed [see Figure S1F from *Baubec et al. (2013)*]. Since overexpression of proteins can promote promiscuous binding as assayed by ChIP (*Baresic et al., 2014*; *Fernandez et al., 2003*), it is possible some of the peaks of Mbd3 binding identified by this method are non-physiological. Second, all Mbd3 isoforms were immunoprecipitated in the Yildirim study, rather than the single largest isoform examined in Baubec et al. Third, whereas Yildirim et al. examined Mbd3 binding to promoter-proximal regions of genes, Baubec et al. focused mainly on a subset of LMRs and ICPs. Regardless of these differences, comprehensive analysis of the data available from Baubec et al. revealed a strong reduction in Mbd3 binding in TKO cells at the majority of bio-Mbd3 binding sites (*Figure 1B–C*). Similarly, our re-analysis of the bio-Mbd2 ChIP-seq data also revealed a strong reliance on DNA methylation (*Figure 1D–E*).

To more thoroughly address the role of DNA methylation in the regulation of Mbd3 and Mbd2 binding in ES cells, we performed a series of ChIP-seq experiments for endogenous Mbd3 and Mbd2 in *Dnmt1* or *Tet1* KD cells. Duplicate ChIP-seq experiments performed with polyclonal antibodies targeting the endogenous MBD proteins (*Figure 2*), independent triplicate ChIP-qPCR experiments performed with the same polyclonal antibodies in *Dnmt1* KD, *Tet1* KD, or *Dnmt1* KO cells (*Figure 2—figure supplement 2*), and ChIP experiments performed on endogenously C-terminal FLAG-tagged versions of Mbd2 or Mbd3 (*Figure 2—figure supplement 3*) all yielded the same result: depletion of *Dnmt1* or *Tet1* results in reduced binding of Mbd3 and Mbd2 throughout the genome. Furthermore, experiments utilizing point mutations that eliminate the catalytic activity of Tet1 demonstrated the DNA demethylase activity of Tet1 is necessary for Mbd3 and Mbd2 binding (*Figure 3*) and (along with data from *Dnmt1* KO cells) show that our findings are not an artifact of potential off-target effects from RNAi. These data, together with a re-evaluation of published data, conclusively demonstrates the requirement for DNA methylation/hydroxymethylation for normal chromatin binding by Mbd3 and Mbd2 in ES cells.

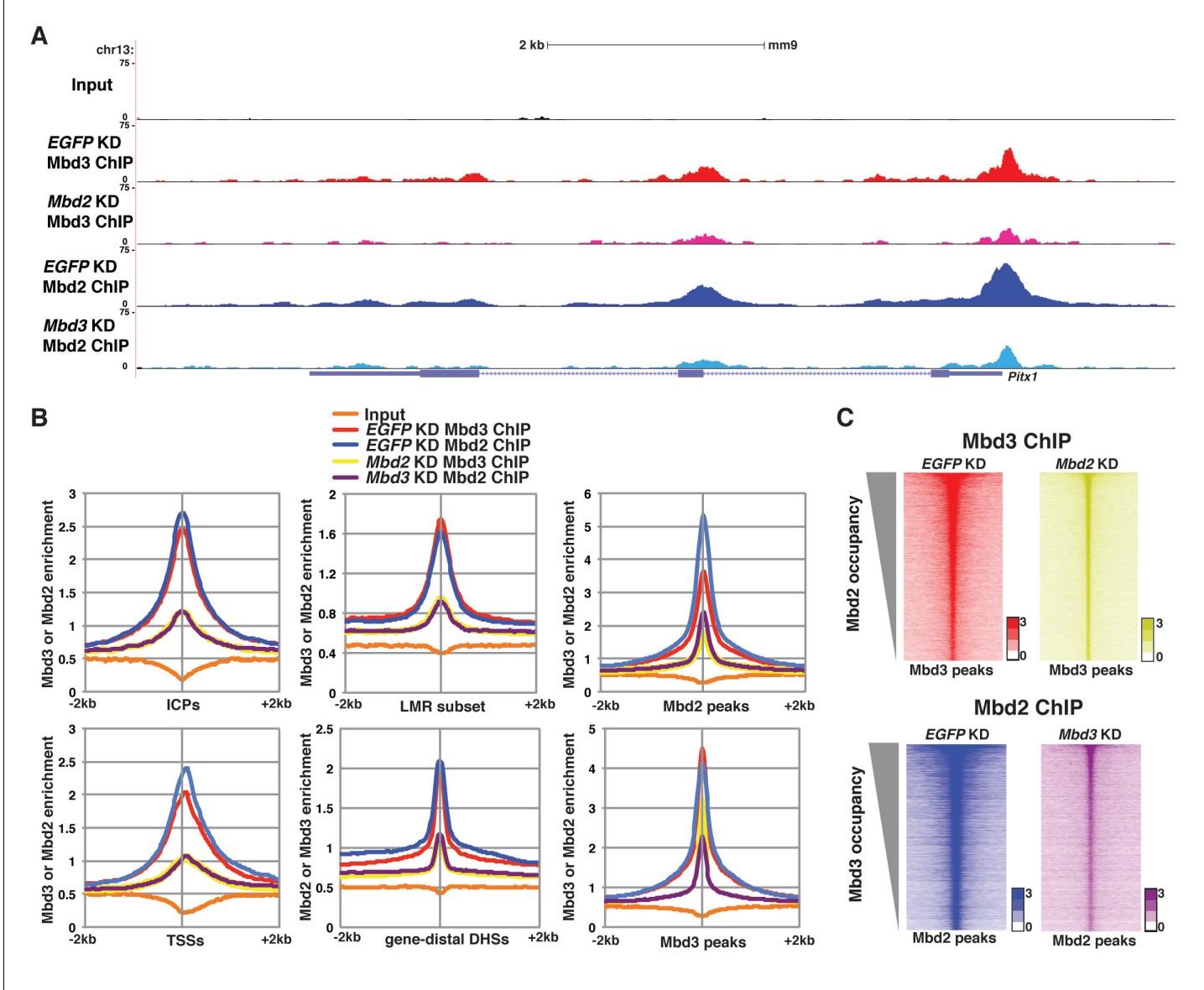

**Figure 5.** Mbd3 and Mbd2 are required for each other's binding in ES cells. (**A**) Genome browser tracks of ChIP-seq experiments examining Mbd3 or Mbd2 occupancy in control (*EGFP* KD), *Mbd2* KD, or *Mbd3* KD ES cells over one example locus (*Pitx1*). (**B**) Aggregation plots of Mbd3 or Mbd2 ChIP-seq data showing occupancy over ICPs (top left panel), annotated TSSs (bottom left panel), LMR subset (top middle panel), gene-distal DHSs (bottom middle panel), Mbd2 peaks (top right panel), and Mbd3 peaks (bottom right panel) ± 2 kb in control (*EGFP* KD), *Mbd3* KD, or *Mbd2* KD ES cells. (**C**) Heatmaps of Mbd3 enrichment over Mbd3 binding sites ± 2 kb sorted by Mbd2 occupancy (top panel) and Mbd2 enrichment over Mbd2 binding sites ± 2 kb sorted by Mbd3 occupancy (bottom panel) in control (*EGFP* KD), *Mbd2* KD, or *Mbd3* KD ES cells. The profiles shown in the browser tracks, aggregation plots, and heatmaps represent the average of two biological replicates.

The following figure supplements are available for figure 5:

**Figure supplement 1.** Interplay between Mbd3 and Mbd2.

**Figure supplement 2.** Mbd3 is required for Dnmt1 occupancy in ES cells.

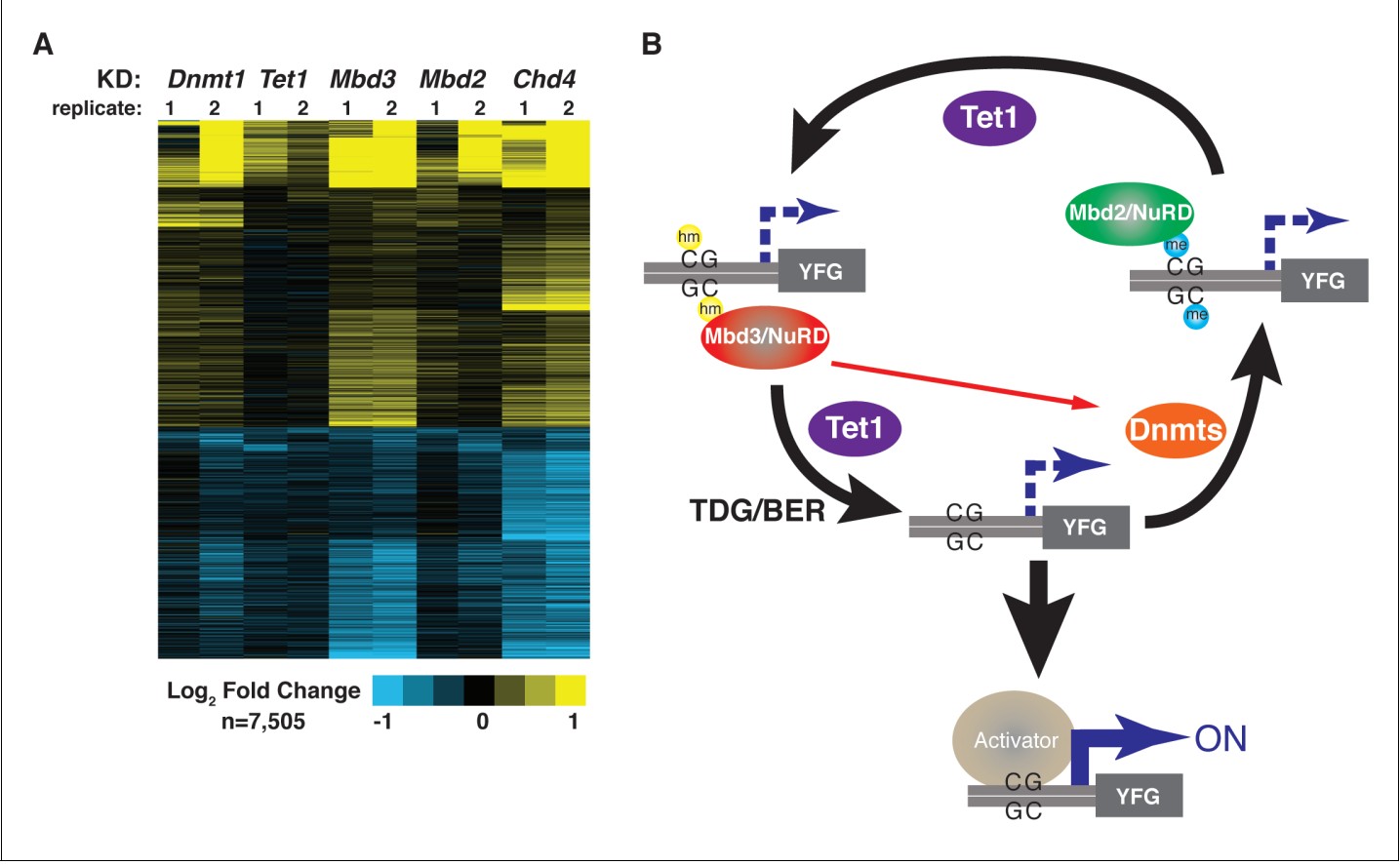

**Figure 6.** KD of DNA methylation/hydroxymethylation machinery or its readers results in similar effects on gene expression. (**A**) K-means clustering (k = 3) of genes misregulated (adjusted p<0.05) upon *Dnmt1* KD, *Tet1* KD, *Mbd3* KD, *Mbd2* KD, or *Chd4* KD. Upregulated genes are indicated in yellow and downregulated genes are indicated in blue. (**B**) Model for interdependent regulatory mechanism mediated by DNA methylation/hydroxymethylation and readers of 5mC/5hmC. The DNA methylation/hydroxymethylation machinery (Dnmt1 and Tet1) and its readers (Mbd3 and Mbd2) form a regulatory loop where disruption of one factor results in altered methylation patterns and MBD binding. Binding of a gene activator to unmethylated site, loss of methylation by another mechanism, or loss of a reader of 5mC/5hmC can break the regulatory loop. YFG (your favorite gene) indicates a generic gene body.

The following figure supplement is available for figure 6:

**Figure 6—figure supplement 1.** *Dnmt1* KD, *Tet1* KD, *Mbd3* KD, *Mbd2* KD, or *Chd4* KD have correlated effects on gene expression.

Having confirmed a role for cytosine methylation in Mbd3 and Mbd2 binding to DNA in ES cells, we sought to further investigate the mechanism through which DNA methylation regulates binding of these factors. Previous studies revealed that Mbd3 is required for 5mC at some locations in ES cells (*Latos et al., 2012*), but may prevent methylation at other sites (*Cui and Irudayaraj, 2015*). Our data examining 5mC and 5hmC enrichment genome-wide reveals a decrease in global 5mC and 5hmC levels in ES cells depleted of Mbd3 or Mbd2 (*Figure 4*). This decrease is likely due to the reduction in Dnmt1 binding throughout the genome we observed in *Mbd3* KO cells (*Figure 5—figure supplement 2*), consistent with findings in cancer cell lines (*Cai et al., 2014*; *Choi et al., 2013*; *Morey et al., 2008*), although other mechanisms may also contribute to this phenomenon.

Consistent with previous studies (*Günther et al., 2013*), we found that Mbd3 and Mbd2 binding overlap at many locations. This observation, along with the fact that *Mbd3* is necessary for normal DNA methylation, suggested *Mbd3* might be indirectly necessary for chromatin binding by Mbd2. We therefore investigated the potential interplay between these factors and found that Mbd2 requires *Mbd3* for binding, and vice versa, throughout the genome (*Figure 5*). Together these data support a model in which Mbd3 and Mbd2 rely on DNA methylation and hydroxymethylation for

binding, while these factors are also required for DNA methylation and hydroxymethylation. Such a model must be consistent with the following observations: (1) DNA methylation is necessary for proper Mbd2 and Mbd3 localization (as shown by *Dnmt1* KD and KO experiments described in this study). (2) DNA demethylation is required for proper Mbd2 and Mbd3 localization (*Tet1* KD and *Tet1c.i.* studies here, and data from *Yildirim et al. (2011)*). (3) Mbd2 binds DNA harboring 5mC (*Hendrich and Bird, 1998*; *Zhang et al., 1999*) and Mbd3 binds DNA binding 5hmC much more strongly than DNA containing 5mC (*Yildirim et al., 2011*). (4) *Mbd2* and *Mbd3* are required for normal levels of both 5mC and 5hmC (MeDIP/hMeDIP, TLC, and other experiments described in this study). (5) Dnmt1 localization depends on *Mbd3* (ChIP studies described in this study). (6) RNAi-mediated depletion of *Mbd2*, *Mbd3*, *Dnmt1*, *Tet1*, or *Chd4* results in misregulation of many of the same genes (expression profiling described in this study). Therefore, we propose that these two MBD proteins and DNA modifications all contribute to a regulatory loop that modulates gene expression (*Figure 6B*). Consequently, upon disruption of any component of this regulatory loop (through depletion, KO, or mutation of the catalytic domain), the gene regulatory outcome is similar in ES cells: binding of both NuRD complexes is reduced, altering gene expression. While the molecular mechanisms underlying these interdependencies are unclear, physical interactions between Mbd3 and DNA methylation/hydroxymethylation proteins have previously been observed (*Cai et al., 2014*; *Yildirim et al., 2011*). Furthermore, predictive studies utilizing published ChIP-seq datasets suggest interdependency between the Mbd2 and Tet1 proteins in ES cells (*Juan et al., 2016*).

How widespread are the interdependencies among Mbd2, Mbd3, and the DNA modification machinery? *Mbd2* KO mice are viable and fertile, whereas *Mbd3* KO mice are lethal (*Hendrich et al., 2001*; *Kaji et al., 2006*; *Reynolds et al., 2012*). *Mbd3* is required for ES cell differentiation, suggesting Mbd3 may be sufficiently functional in the absence of Mbd2 for its essential roles in vivo. Consistent with this possibility, *Mbd3* KD had a stronger effect on some shared Mbd2/Mbd3 target genes than *Mbd2* KD in ES cells. Alternatively, Mbd2/Mbd3 interdependence may be lost during differentiation, or within specific lineages, which may partially account for the different phenotypes of *Mbd2* and *Mbd3* KO mice. Further studies will be required to distinguish between these possibilities.

## Materials and methods

### Cell culture

Mouse ES cells were derived from E14 (*Hooper et al., 1987* RRID:CVCL_C320) and were cultured as previously described (*Chen et al., 2013*). Cells have been verified that they are of male mouse origin through sequencing performed in this and previous studies, and were previously tested to ensure they were free of mycoplasma. RNAi-mediated KD was performed for 48 hr with endoribonuclease III digested siRNAs (esiRNAs) as previously described (*Fazzio et al., 2008*). The *Mbd3abc-3XFLAG* cell line was described previously (*Yildirim et al., 2011*).

### Generation of cell lines

To generate the *Mbd3a-3XFLAG* lines, the mouse *Mbd3a* coding sequence was amplified from cDNA, a 6XHis + 3 XFLAG tag was added to the 3' end, and it was cloned into pLJM1-*EGFP* (courtesy of David Sabatini's lab; Addgene plasmid # 19319) at the AfeI and BstBI sites, replacing the *EGFP* gene. 293T cells were transfected with pLJM1-*Mbd3a-3XFLAG* and Lenti-X plasmids (Clontech, Mountain View, CA, USA) via X-tremeGENE (Roche, Branford, CT, USA) to produce lentivirus. Virus was collected three times between 40 and 64 hr and concentrated as described previously (*Chen et al., 2013*). Virus was introduced to E14 ES cells with 8 µg/mL hexadimethrine bromide (Polybrene; Sigma, St. Louis, MO, USA). Cells were selected with puromycin, diluted to single clones, and screened for *Mbd3a-3XFLAG* expression by western blotting of whole cell lysates.

To generate the endogenously tagged *Mbd2-3XFLAG* lines, sequence corresponding to *Mbd2* exon six was synthesized (IDT, Coralville, IA, USA) with a 6XHis + 3 XFLAG tag before the stop codon and cloned into pBluescript before an internal ribosome entry site followed by a NeoR gene and polyadenylation site, with ~2 kb of homology to either side of the *Mbd2* target site, which was amplified from genomic DNA. Oligonucleotides (F: CACCGTGAGGCGTAAGAATATGATC, R: AAACGATCATATTCTTACGCCTCAC) were cloned into pX330-puroR (*Cong et al., 2013*) for

expression of a guide RNA that targets Cas9 to the C-terminus of *Mbd2*, in order to increase efficiency of targeted homologous recombination with the homology template. The guide RNA was designed over the stop codon of the gene so that it could not target the homology construct or a recombined *Mbd2-3XFLAG*. The two plasmids were transfected together into E14 ES cells by electroporation, and the cells were subsequently selected with G418. Clones were screened by PCR on genomic DNA and by Western blot for FLAG-tagged Mbd2 on whole cell lysates.

*Dnmt1* and *Mbd3abc* knockout (KO) lines were created by CRISPR/Cas9 targeted double-strand breaks and error-prone DNA repair to generate frameshift mutations (*Cong et al., 2013*; *Wang et al., 2013*). Oligonucleotides for guide RNAs specific to *Dnmt1* exons 15 (F: CACCGGG TTAGGGTCGTCTAGGTGC, R: AAACGCACCTAGACGACCCTAACCC) or 20 (F: CACCGAAAC TGGGCGTGGCGTAAGA, R: AAACTCTTACGCCACGCCCAGTTTC) or to *Mbd3* exon 5 (F: CACCGGTGTGTAGAGCACTCGCAA, R: AAACTTGCGAGTGCTCTACACACC) were cloned into pX330-puro. The plasmids were separately transfected into E14 ES cells by electroporation (for *Dnmt1* KO) or using FuGENE HD (Promega, Madison, WI, USA, for *Mbd3abc* KO) and clones were selected with puromycin. Candidates were screened by Western blotting of Dnmt1 or Mbd3 protein from whole cell lysates and by sequencing of PCRs on genomic DNA at the target sites.

*Tet1* catalytically inactive (*Tet1c.i.*) lines were generated by CRISPR/Cas9 targeted homologous recombination. Oligonucleotides (F: CACCGATTTTTGTGCCCATTCTCACA R: AAACTGTGAGAA TGGGCACAAAAATC) were cloned into pX330-puro for a guide RNA that targets the *Tet1* catalytic site for Cas9 cleavage. A sequence harboring the *Tet1* H1652Y and D1654A mutations (previously demonstrated to eliminate Tet1 catalytic activity (*Ito et al., 2010*; *Tahiliani et al., 2009*)) was synthesized (IDT) with ~500 bp of homology to the genomic DNA on each side and cloned into pCR2.1 (Invitrogen, Grand Island, NY, USA). The two plasmids were transiently transfected together into E14 ES cells with FuGENE HD (Promega), and the cells were subsequently puromycin selected for ~40 hr to enrich for transfected cells. Clones were screened for incorporation of the mutations at both alleles by specific digestion by PsiI (NEB, Ipswich, MA, USA) of PCRs from genomic DNA. PCR products were sequenced. Western blots of whole cell lysates from the lines were performed to ensure that mutant Tet1 protein levels were unchanged relative to wild-type levels.

## RT-qPCR

RNA was isolated from cells using TRIzol (Invitrogen) and used to synthesize cDNA with random hexamers (Promega) as described (*Hainer et al., 2015*). cDNA was used in quantitative PCR reactions with *Dnmt1*, *Tet1*, *Mbd3*, *Mbd2*, *Chd4*, or *GAPDH* specific primers (see *Table 2*) and a FAST SYBR mix (KAPA Biosystems, Woburn, MA, USA) on an Eppendorf Realplex.

**Table 2.** Related to *Figure 2—figure supplement 1–3*. Primer list for qPCR.

| Gene | Forward | Reverse |
|------|---------|---------|
| *Dnmt1* | TGTTCTGTCGTCTGCAACCT | GCCATCTCTTTCCAAGTCTTT |
| *Tet1* | TCACAGGCACAGGTTACAAAAG | TCCTTACATTTTCAAGGGGATG |
| *Mbd3* | GGCCACAGGGATGTCTTTTACT | CTTGACCTGGTTGGAAGAATCA |
| *Mbd2* | AACCAAATTCACGAACCACC | CCTTGTAGCCTCTTCTCCCA |
| *Chd4* | AAGTTTGCAGAGATGGAAGAGC | GGTCGTAGTCCTGAATCTCCAC |
| *Farp1* | AACTGCAAGTCATTCTAAATCTCG | GGTATTCAATGCCAGAGACACA |
| *Ppp2r2c* | CGAATTATCCAGCTCTGCCTTA | TGGAGGAGAGACTTAGGGGTGT |
| *Gpr83* | GAGCCACCTTACTGTAGGGAATG | CACGCTCACCAGCTTTCTGTA |
| *Cldn7* | ACCTTTGGAAGAGCAGTCAGTG | CCTTCTCCATCCACACACTTTC |
| *Tgfb1* | AAGTCAGAGACGTGGGGACTTCTTG | AGTCTTCGCGGGAGGCGGGGT |
| *Trh* | TAATGCCTCTGACCTGGGATC | CCCACATCCTAATTCCAAAGTG |

**Table 3.** Related to *Figure 6—figure supplement 1*. Correlation coefficients for pairwise combinations of factor KD. *Tip60* KD (a histone acetyltransferase with repressive functions in ES cells) and *Brg1* KD (an ATP-dependent chromatin remodeling enzyme) are included for comparison.

| Dnmt1 | Tet1 | Mbd3 | Mbd2 | Chd4 | Tip60 | Brg1 | KD: |
|---|---|---|---|---|---|---|---|
| 1.0 | 0.508 | 0.693 | 0.612 | 0.645 | -0.033 | 0.051 | Dnmt1 |
| | 1.0 | 0.441 | 0.551 | 0.475 | -0.003 | 0.104 | Tet1 |
| | | 1.0 | 0.712 | 0.831 | -0.101 | 0.308 | Mbd3 |
| | | | 1.0 | 0.656 | -0.027 | 0.033 | Mbd2 |
| | | | | 1.0 | -0.094 | 0.269 | Chd4 |
| | | | | | 1.0 | 0.199 | Tip60 |
| | | | | | | 1.0 | Brg1 |

## Western blotting

Western blotting was performed as previously described (*Hainer et al., 2015*). From whole cell lysates, 30 μg of protein were separated by SDS-PAGE, transferred to nitrocellulose (Life Sciences, St Petersburg, FL, USA), and assayed by immunoblotting. The antibodies used to detect proteins were: anti-Dnmt1 (1:1000, Sigma D4692, RRID:AB_262096), anti-Tet1 (1:1000, Millipore, Billerica, MA, USA, 09–872, RRID:AB_10806199), anti-Mbd3 (1:1000, Bethyl Labs, Montgomery, TX, USA, A302-529A; RRID:AB_1998976), anti-Mbd2 (1:1000, Bethyl Labs A301-632A, RRID:AB_1211478), anti-Chd4 (1:1000, Bethyl Labs A301-082A, RRID:AB_873002), anti-FLAG M2 (1:10,000, Sigma F1804, RRID:AB_262044) and anti-β-actin (1:50,000, Sigma A1978, RRID:AB_476692).

## Chromatin immunoprecipitation (ChIP)-qPCR and ChIP-seq

ChIP experiments were performed as previously described (*Hainer et al., 2015*). Ten million cells were fixed, washed with ice-cold PBS, pelleted, lysed through sonication on high in a Bioruptor UCD-200 (Diagenode, Delville, NJ, USA) in SDS Lysis Buffer (1% SDS, 10 mM EDTA, 50 mM Tris-HCl pH 8.0), and supernatants were saved. Input samples were stored (30 μL) at 4°C while the remainder of the sheared chromatin was combined with antibody-coupled protein A magnetic beads (NEB) and incubated at 4°C overnight with constant rotation. Mbd3 antibody (Bethyl Labs A302-529A, RRID:AB_1998976), Mbd2 antibody (Bethyl Labs A301-632A, RRID:AB_1211478), or Dnmt1 antibody (Sigma D4692, RRID:AB_262096) coupled protein A magnetic beads (NEB) were blocked with 5 mg/mL BSA overnight at 4°C, prior to incubation with sheared chromatin. Magnetic beads were washed, material was eluted at 65°C on a thermomixer, and the eluent was incubated at 65°C overnight to reverse crosslinking. Input DNA was diluted with 170 μL elution buffer (20 mM Tris-HCl pH 8.0, 100 mM NaCl, 20 mM EDTA, 1% SDS) and treated similarly. Samples were treated with RNaseA/T1 (Ambion, Carlsbad, CA, USA) followed by proteinase K (Ambion) and then PCI extracted. Ethanol precipitated ChIP-enriched DNA was then used for library construction or as a template for qPCR reactions.

FLAG-ChIP experiments were performed as previously described (*Chen et al., 2013*). Cells were fixed, washed with PBS, and pelleted. After resuspension and incubation with Lysis buffer 1 (50 mM HEPES KOH pH 7.6, 140 mM NaC, 1 mM EDTA, 10% (w/v) glycerol, 0.5% NP-40, 0.25% Triton X100), pellets were sheared in Lysis buffer 2 (10 mM Tris-HCl pH 8.0, 200 mM NaCl, 1 mM EDTA, 0.5 mM EGTA) for sonication in a Bioruptor. Sheared chromatin was incubated at 4°C overnight with protein G magnetic beads (NEB) pre-blocked and coupled with anti-Flag M2 (Sigma F1804, RRID: AB_262044). Magnetic beads were washed, material was eluted at 65°C on a thermomixer, and the eluent was incubated at 65°C overnight to reverse crosslinking. Input DNA was diluted with 170 μL elution buffer and treated similarly. Samples were treated with RNaseA/T1 (Ambion) followed by proteinase K (Ambion) and then PCI extracted. Ethanol precipitated, ChIP-enriched DNA was then used for library construction or as a template for qPCR reactions.

## Library construction

Libraries of ChIP-enriched DNA were prepared as described previously (*Chen et al., 2013*). Samples were end-repaired, A-tailed, and adaptor-ligated with DNA purification over a column between each step. DNA was PCR amplified with KAPA HiFi polymerase using 16 cycles of PCR. Each library was size-selected on a 1% agarose gel, its concentration determined, and the integrity was confirmed by sequencing ~10 fragments from each library. Libraries were sequenced on an Illumina HiSeq2000 using single-end sequencing at the UMass Medical School deep sequencing core facility.

## Data analysis

Single-end fastq reads were split by barcode adapter sequences, adapter sequences were removed, and reads were mapped to the mm9 genome using bowtie (RRID:SCR_005476), allowing up to three mismatches. Aligned reads were processed in HOMER (*Heinz et al., 2010*, RRID:SCR_010881). Genome browser tracks were generated from mapped reads using the 'makeUCSCfile' command. Peaks were called using the 'findPeaks' command after reads were mapped. Peaks were called individually from replicate datasets and those peaks enriched in both datasets were retained. Mapped reads were aligned over specific regions using the 'annotatePeaks' command to make 20 bp bins over regions of interest and sum the reads within each window. Experiments were aligned over the following datasets: TSS reference sites (from HOMER software), intermediate CpG promoters (ICPs, defined in (*Weber et al., 2007*), low methylated regions (LMRs from (*Stadler et al., 2011*) with or without additional selection criteria used by Baubec et al. – exclusion of LMRs under 150 bp and within 3 kb of other LMRs, referred to as 'LMR subset'), DNaseI hypersensitive sites (DHSs) from mouse ENCODE data (GSM1014154) with TSS locations removed, Mbd3 peaks (peaks called from both *EGFP* KD Mbd3 ChIP-seq libraries using the 'findPeaks' command in HOMER with input as a control) and Mbd2 peaks (peaks called from both *EGFP* KD Mbd2 ChIP-seq libraries using the 'findPeaks' command in HOMER with input as a control). After anchoring mapped reads over reference sites, aggregation plots were generated by averaging data obtained from biological replicates. Overlapping peaks for *Figure 2B* were identified using the 'mergePeaks' command.

Analysis of data from (*Baubec et al., 2013*) was performed similarly. Data were downloaded from GSE39610 (see *Table 1* for SRA numbers) and converted to fastq files using SRA dump. Barcodes were trimmed where necessary, and reads were mapped to the mm9 genome using bowtie (RRID: SCR_005476), allowing up to three mismatches. Aligned reads were processed in HOMER (*Heinz et al., 2010*, RRID:SCR_010881) and analysis was confirmed independently using R and Macs2. Peaks for bio-Mbd2, bio-Mbd3, bio-Mbd1a, bio-Mbd1b, bio-Mbd4, or bio-MeCP2 were called using the 'findPeaks' command after reads were mapped and separately called using Macs2 to confirm validity of called peaks. Mapped reads were aligned over specific regions using the 'annotatePeaks' command to make 20 bp bins over regions of interest and sum the reads within each window. Experiments were aligned over the same genomic features described above and bio-Mbd3 peaks (called from bio-Mbd3 ChIP-seq in WT cells using 'findPeaks' command in HOMER with pooled input as a control), bio-Mbd2 peaks (called from bio-Mbd2 ChIP-seq in WT cells using 'findPeaks' command in HOMER with pooled input as a control), bio-Mbd1a peaks (called from bio-Mbd1a ChIP-seq in WT cells using 'findPeaks' command in HOMER with pooled input as a control), bio-Mbd1b peaks (called from bio-Mbd1b ChIP-seq in WT cells using 'findPeaks' command in HOMER with pooled input as a control), bio-Mbd4 peaks (called from bio-Mbd4 ChIP-seq in WT cells using 'findPeaks' command in HOMER with pooled input as a control), or bio-MeCP2 peaks (called from bio-MeCP2 ChIP-seq in WT cells using 'findPeaks' command in HOMER with pooled input as a control). To confirm this analysis, bio-Mbd2 and bio-Mbd3 data were downloaded and mapped independently and mapped reads were annotated and summarized using the Bioconductor package ChIPpeakAnno (*Zhu et al., 2010*; *Zhu, 2013*, RRID:SCR_012828). Furthermore, bio-Mbd2 and bio-Mbd3 peaks were confirmed independently by calling peaks from WT bio-Mbd2 or WT bio-Mbd3 ChIP-seq experiments, respectively, using Macs2. Finally, we re-analyzed a subset of these data after re-aligning reads (using bowtie, RRID:SCR_005476) while allowing only two mismatches instead of three. The use of two or three mismatches during alignment had no effect on the results.

Analysis of LMRs was performed as follows. LMR locations were downloaded from *Stadler et al. (2011)* (referred throughout as 'total LMRs'). As per the description in *Baubec et al. (2013)*, LMRs < 150 bp and within 3 kb of other LMRs or unmethylated regions were removed (referred

throughout as 'LMR subset'). The LMRs not included in this subset are referred throughout as 'removed LMRs'. To assess the genomic locations of these LMRs, the peaks were annotated in HOMER (*Heinz et al., 2010*, RRID:SCR_010881) using the 'annotatePeaks' command.

Mbd3 ChIP-seq data from (*Yildirim et al., 2011*) was downloaded and converted to fastq using SRA-dump from GSE31690. Reads were mapped to the mm9 genome using bowtie, allowing up to three mismatches. Aligned reads were processed in HOMER (*Heinz et al., 2010*). Mapped reads were aligned over gene-distal DHSs. Replicates for Mbd3 ChIP-seq in control cells were examined and resulted in similar profiles. Only replicate 2 (which has higher read coverage) is shown in *Figure 2—figure supplement 4E*.

## Methylated and hydroxymethylated DNA immunoprecipitation sequencing (MeDIP-seq and hMeDIP-seq)

MeDIPs and hMeDIPs were performed as previously described (*Mohn et al., 2009*) with slight modifications. Genomic DNA was isolated from ES cells by resuspending cells in ES cell lysis buffer (10 mM Tris-HCl pH 7.5, 10 mM EDTA, 10 mM NaCL, 0.5% sarkosyl), treating with RNase A/T1 (Ambion), and incubating with 1 µg/µL proteinase K overnight at 55°C. DNA was cleaned through PCI extraction and isopropanol precipitation. Genomic DNA was then diluted to 20 ug/mL in TE buffer and sonicated in a Bioruptor UCD-200 (Diagenode) on low for 20 min with intervals of 15 s on/15 s off at 4°C. Sheared DNA (ranging in size from 200 to 600 bp) was prepared for deep sequencing and further processed by end-repairing, A-tailing, and adaptor-ligating as previously described for MNase-Seq libraries (*Hainer et al., 2015*) with DNA purification through PCI extraction and ethanol precipitation between each step. Of the prepared DNA, 1.5 µg was diluted in TE buffer up to 450 µL, denatured for 10 min at 95°C and immediately transferred to ice for 10 min. Of the prepared DNA, 1.5 µg was stored for input samples. Denatured DNA was then incubated with 50 µL 10X DNA IP buffer (100 mM Na-HPO$_4$ pH 7.0, 1.4 M NaCl, 0.5% Triton X-100) and either 1.5 µL anti-5mC (Eurogentec Fremont, CA, USA, BI-MECY, RRID:AB_2616058) or 1.5 µL anti-5hmC (Active Motif, Carlsbad, CA, USA, 39791, RRID:AB_2630381) overnight at 4°C with constant rotation. 40 µL of magnetic beads (anti-mouse IgG for 5mC and anti-rabbit IgG for 5hmC, Life Technologies, Waltham, MA, USA) were pre-blocked with BSA and then resuspended in 1X DNA IP buffer. Beads were then added to the DNA/antibody mixture and incubated for 2 hr at 4°C with constant rotation. Beads were then washed three times with 700 µL 1X DNA IP buffer, and DNA was eluted from the beads by incubating at 50°C on a thermomixer with 250 µL 1X DNA IP buffer and 4 µL proteinase K (20 µg/µL) for 3 hr. Input samples were diluted up to 250 µL with DNA IP buffer and treated similarly. DNA from both input and IP samples were cleaned through PCI extraction and ethanol precipitation. DNA was PCR amplified with KAPA HiFi polymerase using 16 cycles of PCR. Each library was size-selected on a 1% agarose gel, its concentration was determined, and the integrity was confirmed by TOPO-cloning (Invitrogen) a portion and sequencing ~10 fragments from each library. Libraries were sequenced on an Illumina HiSeq2000 using single-end sequencing at the UMass Medical School deep sequencing core facility. Resulting fastq sequences were processed and analyzed as described for ChIP-seq libraries using the HOMER software (*Heinz et al., 2010*, RRID:SCR_010881).

## Dot blotting

Twofold serial dilutions of sheared genomic DNA (200–400 bp) starting at 300 ng were denatured at 95°C for 10 min, then put on ice immediately for 10 min. Denatured DNA samples were spotted onto Amersham Hybond N+ nylon membrane (GE Healthcare, Uppsala, Sweden) and the membranes were UV crosslinked. For 5mC, the membrane was incubated in 0.1% SDS overnight, washed five times with PBS-T, blocked with 5% nonfat milk and 3% BSA for 4 hr, washed three times with PBS-T, incubated with anti-5mC (1:1000, Eurogentec BI-MECY, RRID:AB_2616058) for 1 hr, washed three times with PBS-T, incubated with HRP conjugated anti-mouse secondary (1:10,000, Bio-Rad, Hercules, CA, USA, 170–6516, RRID:AB_11125547) for 1 hr, washed three times with PBS-T, and detected with enhanced chemiluminescence. For 5hmC, the membrane was blocked with 5% nonfat milk for 4 hr, washed three times with PBS-T, incubated overnight with anti-5hmC (1:1000, Active Motif 39791, RRID:AB_2630381), washed three times with PBS-T, incubated with HRP conjugated anti-rabbit secondary (1:10,000, Bio-Rad 170–6515, RRID:AB_11125142) for 1 hr, washed three times with PBS-T, and detected with enhanced chemiluminescece. For loading, sheared gDNA samples

were diluted simultaneously, spotted directly onto Amersham Hybond N+ nylon membrane (GE Healthcare) and the membranes were UV crosslinked. Membranes were incubated with 0.2% methylene blue for 5 min and washed five times with water.

## Thin layer chromatography (TLC)

TLC was performed as previously described (*Ficz et al., 2011*; *Tahiliani et al., 2009*). 2 µg of genomic DNA isolated from control, *Dnmt1* KD, *Mbd3* KD, or *Mbd2* KD ES cells was digested overnight with 50 units of MspI (NEB) and 10 µg of RNaseA/T1 (Ambion) at 37°C. The reaction was then heat inactivated at 65°C, DNA was dephosphorylated with 10 units of rSAP (NEB) for 1 hr at 37°C and cleaned over a DNA clean and concentrator column (Zymo Research, Irvine, CA, USA). 400 ng of DNA was incubated with 10 µCi $^{32}$P-ATP and 10 U of T4 PNK for 1 hr at 37°C. Labeled DNA was precipitated, resuspended in 18 µL 30 mM Tris pH 8.9, 15 mM MgCl$_2$, 2 mM CaCl, and fragmented to single nucleotides with 10 units of DNaseI (NEB) and 10 µg SVPD (Worthington, Lakewood, NJ, USA) for 3 hr at 37°C. 3 µL of samples were spotted onto PEI cellulose F TLC plates (Millipore) and developed in isobutyric acid:H$_2$O:NH$_3$ (66:20:1 v/v/v) for 15 hr. Plates were dried and exposed to film for 30 hr. Plate was then exposed to a phosphorimager screen for 110 hr and scanned on a Typhoon FLA 700 (GE Healthcare). Pixels were quantitated using Fiji ImageJ software.

## Methylation-sensitive restriction digests

Five micrograms of genomic DNA isolated from control, *Dnmt1* KD, *Mbd3* KD, or *Mbd2* KD ES cells was digested at 37°C for 12 hr with 50 units of MspI or HpaII (both from NEB) followed by 20 min of heat inactivation at 65°C. MspI and HpaII recognize the 4 bp sequence CCGG, but HpaII is blocked by CpG methylation whereas MspI is insensitive to CpG methylation. 15 µL of the 50 µL reaction was run on a 1% agarose gel to visualize the cutting efficiency of each enzyme.

## Microarray analysis

Microarray analysis was performed as previously described (*Chen et al., 2013*). Total RNA from control (*EGFP* KD), *Dnmt1* KD, *Tet1* KD, *Mbd3* KD, *Mbd2* KD, or *Chd4* KD ES cells was subjected to RNA amplification and labeling using the Low Input Quick Amp Labeling Kit (Agilent, Santa Clara, CA, USA). Following this procedure, 50 picomoles of cRNA was used for fragmentation and hybridization on Agilent 4 × 44K mouse whole-genome microarrays. Slides were scanned on a DNA microarray scanner (Agilent G2565CA), and fluorescence data were obtained using Agilent Feature Extraction software. The expression data from biological replicates were fit to a linear model using the Bioconductor package *limma* (*Smyth et al., 2005*; *2003*; *Wettenhall and Smyth, 2004*, RRID: SCR_010943), and analyzed as previously described (*Yildirim et al., 2011*). Datasets were visualized through Java TreeView (*Saldanha, 2004*, RRID:SCR_013503). *Tip60* KD and *Brg1* KD microarray data was downloaded from GSE31008 (*Yildirim et al., 2011*) and compared to averaged microarray data for fold change of *Dnmt1* KD, *Tet1* KD, *Mbd3* KD, *Mbd2* KD, and *Chd4* KD. Correlation coefficients for all genes in pairwise factor KD were calculated in R.

## Acknowledgements

We thank I Bach, E Torres, J Benanti, and members of the Fazzio lab for insightful discussions regarding this manuscript. This work was supported by the T32CA130807 training grant and a Leukemia and Lymphoma postdoctoral fellowship to SJH, NIH HD072122 to TGF, and NIH HD080224 to OJR TGF is a Leukemia and Lymphoma Society Scholar. SJH is currently a Special Fellow of the Leukemia and Lymphoma Society. The funders had no role in study design, data collection, analysis, decision to publish, or preparation of the manuscript. All deep sequencing was performed at the UMass Medical School Core facility on a HiSeq2000 supported by 1S10RR027052-01. Genomic datasets have been deposited within GEO (accession number GSE79771). The authors declare no conflict of interest.

## Additional information

### Funding

| Funder | Grant reference number | Author |
|---|---|---|
| Eunice Kennedy Shriver National Institute of Child Health and Human Development | HD072122 | Thomas G Fazzio |
| National Cancer Institute | T32CA130807 | Sarah J Hainer |
| Leukemia and Lymphoma Society | Special Fellows Award | Sarah J Hainer |
| Eunice Kennedy Shriver National Institute of Child Health and Human Development | HD080224 | Oliver J Rando |
| Leukemia and Lymphoma Society | Scholars Award | Thomas G Fazzio |

The funders had no role in study design, data collection and interpretation, or the decision to submit the work for publication.

### Author contributions

SJH, TGF, Conception and design, Acquisition of data, Analysis and interpretation of data, Drafting or revising the article; KNM, Acquisition of data, Contributed unpublished essential data or reagents; JY, LJZ, Analysis and interpretation of data, Drafting or revising the article; L-SE, Drafting or revising the article, Contributed unpublished essential data or reagents; OJR, Conception and design, Drafting or revising the article

### Author ORCIDs

Thomas G Fazzio, http://orcid.org/0000-0002-0353-7466

## Additional files

### Major datasets

The following dataset was generated:

| Author(s) | Year | Dataset title | Dataset URL | Database, license, and accessibility information |
|---|---|---|---|---|
| Hainer SJ, Fazzio TG | 2016 | DNA methylation is required for chromatin binding by Mbd2 and Mbd3 in ES cells | https://www.ncbi.nlm.nih.gov/geo/query/acc.cgi?acc=GSE79771 | Publicly available at the NCBI Gene Expression Omnibus (accession no: GSE79771) |

The following previously published datasets were used:

| Author(s) | Year | Dataset title | Dataset URL | Database, license, and accessibility information |
|---|---|---|---|---|
| Baubec T, Ivanek R, Lienert F, Schübeler D | 2013 | Methylation-dependent and -independent genomic targeting principles of the MBD protein family | https://www.ncbi.nlm.nih.gov/geo/query/acc.cgi?acc=GSE39610 | Publicly available at the NCBI Gene Expression Omnibus (accession no: GSE39610) |
| Yildirim O, Li R, Hung JH, Chen PB, Dong X, Ee LS, Weng Z, Rando OJ, Fazzio TG | 2011 | A key role for Mbd3 in 5-hydroxymethylcytosine-dependent gene regulation in embryonic stem cells | https://www.ncbi.nlm.nih.gov/geo/query/acc.cgi?acc=GSE31690 | Publicly available at the NCBI Gene Expression Omnibus (accession no: GSE31690) |

| Yildirim O, Li R, Hung JH, Chen PB, Dong X, Ee LS, Weng Z, Rando OJ, Fazzio TG | 2011 | A key role for Mbd3 in 5-hydroxymethylcytosine-dependent gene regulation in embryonic stem cells | https://www.ncbi.nlm.nih.gov/geo/query/acc.cgi?acc=GSE31008 | Publicly available at the NCBI Gene Expression Omnibus (accession no: GSE31008) |

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
