## [Decision Letter]

[Editors’ note: a previous version of this study was rejected after peer review, but the authors submitted for reconsideration and the paper was subsequently accepted for publication. The first decision letter after peer review is shown below.]

Thank you for submitting your work entitled "DNA methylation directs genomic localization of Mbd2 and Mbd3 in ES cells" for consideration by *eLife*. Your article has been favorably evaluated by Kevin Struhl (Senior Editor) and three reviewers, one of whom is a member of our Board of Reviewing Editors. Our decision has been reached after consultation between the reviewers. Based on these discussions and the individual reviews below, we regret to inform you that we cannot publish your manuscript in its present form.

All reviewers agree that your manuscript addresses an important area of research, that it adds important information to a high-profile controversy, and that the new experiments and re-analysis of older data are a valuable contribution to the field. As such, your study remains of high interest to *eLife* such that a suitably revised manuscript could be reconsidered. There are a number of important technical concerns that we think might take some additional time to address. One of the most important pertains to Figure 6, as discussed by the first two reviewers. A more objective and quantitative analysis of the microarray expression profile is necessary to make the strong statements about high correlation. Additionally, we agreed that antibody specificity could be an issue and that this study would add positively to the debate if the authors could confirm their IP data in two ways. First, with FLAG ChIP of tagged MBDs to validate their MBD ChIPseq data. Second, with a parallel approach such as TLC to validate their MeDIP and hMeDIP analysis. Finally, all reviewers agreed that the prose could be toned down when comparing the Baubec et al. paper to their previous paper (Yildirim et al.).

*Reviewer #1:*

DNA methylation 'readers' include MBD2, which binds 5-methyl-cytosine, and MBD3, which binds its oxidized form, 5-hydroxymethyl-cytosine. Hainer et al. take a new look at previously published data by Baubec et al. (Cell 153:480, 2013) and perform new experiments, and they cast doubt on the likelihood of DNA methylation-independent binding of MBD2 and MBD3. Further, they find that, surprisingly, MBD2 binding depends on Tet1 gene function, and that MBD2 and MBD3 binding to DNA is mutually dependent.

Overall, this is an important area of research and an area of active debate. The new data provided by the authors, along with their reexamination of the existing data add significantly to the debate. On the other hand, the manuscript does not significantly advance understanding of mechanism. For publication in *eLife*, one might ask the authors to provide a better understanding of MBD function. These points may be open for discussion.

Specific comments in roughly descending order of importance:

Figure 6. The strong assertion is made that gene expression profiles upon knockdown of Dnmt1, Tet1, Mbd3, Mbd2 and Chd4 are all "well-correlated." This seems to be based on visual inspection of the heat maps of microarray results, i.e., the top rows across the different treatments are mostly unregulated genes, and the bottom rows are mostly down regulated. But it seems there should be a more quantitative way to assess patterns of gene regulation. I'd like to see how the expression profiles of Mbd/Tet knockdowns would compare to perturbing a different repressive mechanism, such as knockdown of PRC1 or PRC2 components. Cluster analysis could be done of the columns (different treatments) of the heat map rather than just the rows (genes). You'd expect that, say, profiles of Eed1 and Ezh1 knockdowns would sort together as an "outgroup" compared to the Mbd-Tet-etc treatments.

Mutual dependence of Mbd2, Mbd3, Tet1, 5mC/5hmC: Figure 6 is a nicely done visualization of a possible regulatory cycle that explains the data well. I'd like to see more discussion however of why the different phenotypes vary so much in intact mice for Tet1, Dnmt1, Mbd2 and Mbd3 (e.g. the molecular phenotypes and regulatory cycle here may apply only to ES cells). Does the model predict that the MBDs recruit TET enzymes? Can they be Co-IP'd?

LMR terminology should be very explicit at its first mention, subsection “Unbiased analyses of published data reveal methylation-dependent localization of Mbd3 and Mbd2”, second paragraph. It is confusing later on: in the third paragraph of the aforementioned subsection, and subsection “Loss of Mbd3 or Mbd2 results in reduced 5mC and 5hmC at regulatory regions”. I believe that plain "LMR" is meant strictly in the sense of Baubec et al., 2013, who in turn (I think) use it the sense of Stadler et al., Nature 480:490, 2011, meaning regions >150bp with ~30% methylation. (Does "enrichment of TF binding sites" also apply here?) The authors contrast "LMR" with "total LMRs", which should be defined explicitly. (What is the minimal length? Does the methylations status differ from ~30% etc.)?

Figure 5: light yellow trace for Mbd2 KD Mbd3 ChIP is not visible at all in first two panels ("ICPs" and "LMRs"), and it's hard to see in the other panels. (Presumably it's obscured by brown Mbd3 KD Mbd2 ChIP trace?)

*Reviewer #2:*

Proteins with methyl-binding domains (MBDs) constitute a major class of methylated DNA "readers". The manuscript by Hainer et al. addresses conflicting reports concerning function and binding specificity of two members of that protein family, Mbd2 and Mbd3. There is a disagreement between the findings published in 2011 by the same group and subsequent reports from the Schuebeler lab and Spruijt et al. on the role of CpG methylation and hydroxymethylation in the binding of these two proteins.

This is an intriguing area of research and a high-profile controversy, and the new experiments and re-analysis of the existing data presented by Hainer et al. are a valuable contribution to the field. The authors argue (convincingly, in my view) that many apparent discrepancies are due to differences in study design and analysis. These include particular genomic features analyzed in ChIP experiments, to whether binding specificity was measured using synthetic methylated DNA probes. Among the new experiments, the demonstration that the Tet1 catalytic activity was needed for Mbd2/3 occupancy (Figure 3) in particular served to support authors' argument.

Some of these differences seem highly intriguing but are not pursued any further. For instance, Figure 1 shows that Mbd2 binding in "total LMRs" shows much stronger dependence on methylation than in the subset of LMRs analyzed in Baubec et al. The apparent implication is that most of the methylation dependence is concentrated in those "other" regions which constitute the difference between those two sets. What is special about those regions?

In weighing strengths and weaknesses for *eLife* publication one has to balance the overall quality of the descriptive data presented here with the limited insights into mechanisms and function. I am on the fence as to whether this work presents a sufficiently major advance regarding processes involved in the Mbd2/3 function.

By far the weakest arguments of this paper are presented in Figure 6. Similarity of all expression profiles shown in Figure 6 is not self-evident. It looks like there are some downregulated genes in the top half of the heatmap and upregulated genes in the bottom half, but it's hard to see at the used resolution and color scheme. A more formal analysis of similarity would strengthen the case the authors make. In addition, a change in the heatmap color scheme might help underscore consistency of up- and down-regulated gene sets.

The "regulatory loop" scheme (Figure 6) as formulated in Discussion is strongly overstated. In what sense "the outcome is the same" for disruption of any component of the proposed loop when the phenotypes of Mbd2, Mbd3, and Tet KO ESCs and mice are different? Accordingly, the statement (in the Abstract) that the authors' findings "describe a regulatory loop" is too strong.

*Reviewer #3:*

Summary:

In mammalian cells 5-methylcytosine (5mC) is a well-known heritable epigenetic modification, with well-studied effector proteins. 5-hydroxymethylcytosine (5hmC), initially identified as transient by product of active demethylation process, is less understood. Recent studies show high steady-state levels of 5hmC and effector proteins that preferentially bind to this modification and imply that 5hmC is an abundant form of cytosine modification in mammalian cells. The extent to which Mbd2 and Mbd3 interact with these modifications has been controversial.

In this study the authors investigate the interdependency of Mbd2 and Mbd3 for chromatin localization through binding to DNA methylation and hydroxymethylation sites respectively; they also study how the levels of these two Mbd proteins impact the stability of these two cytosine modifications. The requirement of Tet1 catalytic activity for the localization of Mbd3 and Mbd2 and effect of Mbd3 depletion on Dnmt1 localization are important and interesting findings. The paper offers an advance by providing a good argument that there is strong interdependence between these two binding proteins and the two covalent modifications that they recognize.

I found this paper interesting, but have a few points that are mainly related to helping to resolve the controversies in this area.

1) The interdependence of 5mC and 5hmC levels relative to Mbd2 and Mbd3 was demonstrated with MeDIP and hMeDIP experiments (Figure 4). Since these methods depend on antibody specificity, it would be useful to support these findings with another method such as TLC on overall levels of 5mC and 5hmC.

2) The interdependence of Mbd2 and Mbd3 binding was shown by ChIP with the antibodies that recognize endogenous proteins (Figure 5). Since Mbd2 and Mbd3 have 80% sequence similarity, it would be nice to see the genome-wide effect of Mbd2 and Mbd3 depletion on Mbd3-Flag and Mbd2-Flag occupancy with flag-tag ChIP. This would rule out cross-specificity artifacts.

3) I think the paper would read better if the authors toned down the prose when comparing the Baubec et al. paper to their previous paper (Yildirim et al.) The point they are making is clear, but perhaps too clear as it has a bit of a sledgehammer aspect to it.

4) Figure 5 shows a strong interdependency of Mbd2 and 3. How do the authors explain the phenotypic differences between Mbd3 and Mbd2 null mice?

---

## [Author Response]

[Editors’ note: the author responses to the first round of peer review follow.]

[…]

*Reviewer #1:*

*DNA methylation 'readers' include MBD2, which binds 5-methyl-cytosine, and MBD3, which binds its oxidized form, 5-hydroxymethyl-cytosine. Hainer et al. take a new look at previously published data by Baubec et al. (Cell 153:480, 2013) and perform new experiments, and they cast doubt on the likelihood of DNA methylation-independent binding of MBD2 and MBD3. Further, they find that, surprisingly, MBD2 binding depends on Tet1 gene function, and that MBD2 and MBD3 binding to DNA is mutually dependent.*

*Overall, this is an important area of research and an area of active debate. The new data provided by the authors, along with their reexamination of the existing data add significantly to the debate. On the other hand, the manuscript does not significantly advance understanding of mechanism. For publication in eLife, one might ask the authors to provide a better understanding of MBD function. These points may be open for discussion.*

*Specific comments in roughly descending order of importance:*

*Figure 6. The strong assertion is made that gene expression profiles upon knockdown of Dnmt1, Tet1, Mbd3, Mbd2 and Chd4 are all "well-correlated." This seems to be based on visual inspection of the heat maps of microarray results, i.e., the top rows across the different treatments are mostly unregulated genes, and the bottom rows are mostly down regulated. But it seems there should be a more quantitative way to assess patterns of gene regulation. I'd like to see how the expression profiles of Mbd/Tet knockdowns would compare to perturbing a different repressive mechanism, such as knockdown of PRC1 or PRC2 components. Cluster analysis could be done of the columns (different treatments) of the heat map rather than just the rows (genes). You'd expect that, say, profiles of Eed1 and Ezh1 knockdowns would sort together as an "outgroup" compared to the Mbd-Tet-etc treatments.*

As suggested, in the revised manuscript we quantitatively compare the gene expression profiles of the various KDs and make use of KDs of unrelated factors as outgroups. These analyses show the degree to which the MBD, DNMT, and TET KDs have highly overlapping effects on gene expression, and verify that these alterations are much more similar to each other than the outgroups. The finding that NuRD mediated deacetylation of H3K27 is necessary for PRC2- mediated H3K27 methylation at many loci (Reynolds et al., 2012) complicates use of PRC2 KDs as outgroups. As alternatives, we compared our data with *Tip60* KD, representing a chromatin remodeling complex known to mainly repress gene expression in ES cells (Fazzio et al., 2008), and *Brg1* KD, representing a (mostly) activating chromatin remodeling complex. We downloaded previously published microarray data for both *Tip60* KD and *Brg1* KD (Yildirim et al., 2011) and included these analyses in our revised manuscript (Figure 6—figure supplement 1). This new analysis reveals that gene expression changes in *Tip60* KD or *Brg1* KD ES cells are dissimilar to those of *Dnmt1, Tet1, Mbd3, Mbd2* or *Chd4* KDs. We further show individual KDs of *Dnmt1, Tet1, Mbd3, Mbd2*, and *Chd4* correlate well with each other, but correlate poorly with *Tip60* or *Brg1* KDs (Figure 6—figure supplement 1).

*Mutual dependence of Mbd2, Mbd3, Tet1, 5mC/5hmC: Figure 6 is a nicely done visualization of a possible regulatory cycle that explains the data well. I'd like to see more discussion however of why the different phenotypes vary so much in intact mice for Tet1, Dnmt1, Mbd2 and Mbd3 (e.g. the molecular phenotypes and regulatory cycle here may apply only to ES cells). Does the model predict that the MBDs recruit TET enzymes? Can they be Co-IP'd?*

Previously, we identified a weak interaction between Mbd3/NuRD and Tet1 (Yildirim et al. 2011), which was later confirmed by another group (Shi, Kim et al., 2013). We have attempted co-IP experiments between Mbd2 and Tet1 but have not detected an interaction. We have added discussion of these findings in reference to our model in the fourth paragraph of the Discussion and modified our model to remove the arrows between Mbd2/3 and Tet1 to avoid conflating hypothetical functional connections (that are unnecessary for the model) with connections that are established from data in this manuscript and the published literature. Furthermore, we added more discussion regarding the phenotypes observed in mice and ES cells and their implications with regards to our findings (Discussion, last paragraph).

*LMR terminology should be very explicit at its first mention, subsection “Unbiased analyses of published data reveal methylation-dependent localization of Mbd3 and Mbd2”, second paragraph. It is confusing later on: in the third paragraph of the aforementioned subsection, and subsection “Loss of Mbd3 or Mbd2 results in reduced 5mC and 5hmC at regulatory regions”. I believe that plain "LMR" is meant strictly in the sense of Baubec at et., 2013, who in turn (I think) use it the sense of Stadler et al., Nature 480:490, 2011, meaning regions >150bp with ~30% methylation. (Does "enrichment of TF binding sites" also apply here?) The authors contrast "LMR" with "total LMRs", which should be defined explicitly. (What is the minimal length? Does the methylations status differ from ~30% etc.)?*

In the revised manuscript, we have attempted to clarify the difference by referring to the subset of LMRs examined by Baubec et al. as the “LMR subset” throughout the text. We utilized the same criteria defined by Baubec et al. to generate the LMR subset (remove LMRs smaller than 150bp and those within 3kb of another LMR or a UMR) from the total set of LMRs defined in Stadler et al., 2011 (which we now refer to as “total LMRs”). By the definition of LMRs in Stadler et al., total LMRs have on average 30% methylation, although they range in methylation from 10- 50%. We found that 72% of the LMRs excluded by Baubec et al. (i.e., present within total LMRs but not the LMR subset) were excluded because they are less than 150bp in size, and are therefore smaller than those included in the LMR subset. To address whether these “removed LMRs” are enriched for TF binding sites, we examined the frequencies of 319 known TF motifs in each of the LMR subset, removed LMRs, or total LMR locations. We found on average that the LMR subset had 12.5 motifs per LMR, removed LMRs had 13, and total LMRs had 12.7, suggesting that all three classes of LMRs have similar representation of TF binding sites. Closer examination of the differences between total LMRs and the LMR subset showed that removed LMRs were more highly methylated and hydroxymethylated (compare levels of 5mC and 5hmC in *EGFP* KD for removed LMRs in Figure 4—figure supplement 1 to total LMRs shown in Figure 4—figure supplement 1 and the LMR subset shown in Figure 4; we observe a 5mC peak height of 1.48, 0.96, and 2.64 for total, subset, and removed LMRs, respectively and a 5hmC peak height of 1.51, 1.16, and 2.25 for total, subset, and removed LMRs, respectively). In addition, removed LMRs were more enriched for promoter proximal locations (Figure 4—figure supplement 1 in revised manuscript).

*Figure 5: light yellow trace for Mbd2 KD Mbd3 ChIP is not visible at all in first two panels ("ICPs" and "LMRs"), and it's hard to see in the other panels. (Presumably it's obscured by brown Mbd3 KD Mbd2 ChIP trace?)*

We thank the reviewer for pointing this out. We changed the thickness of the yellow trace to increase its visibility.

*Reviewer #2:*

*Proteins with methyl-binding domains (MBDs) constitute a major class of methylated DNA "readers". The manuscript by Hainer et al. addresses conflicting reports concerning function and binding specificity of two members of that protein family, Mbd2 and Mbd3. There is a disagreement between the findings published in 2011 by the same group and subsequent reports from the Schuebeler lab and Spruijt et al. on the role of CpG methylation and hydroxymethylation in the binding of these two proteins.*

*This is an intriguing area of research and a high-profile controversy, and the new experiments and re-analysis of the existing data presented by Hainer et al. are a valuable contribution to the field. The authors argue (convincingly, in my view) that many apparent discrepancies are due to differences in study design and analysis. These include particular genomic features analyzed in ChIP experiments, to whether binding specificity was measured using synthetic methylated DNA probes. Among the new experiments, the demonstration that the Tet1 catalytic activity was needed for Mbd2/3 occupancy (Figure 3) in particular served to support authors' argument.*

*Some of these differences seem highly intriguing but are not pursued any further. For instance, Figure 1 shows that Mbd2 binding in "total LMRs" shows much stronger dependence on methylation than in the subset of LMRs analyzed in Baubec et al. The apparent implication is that most of the methylation dependence is concentrated in those "other" regions which constitute the difference between those two sets. What is special about those regions?*

As noted in the response to reviewer #1, most of the LMRs excluded by Baubec et al. were excluded due to their size (<150bp), while a smaller portion were excluded due to their location within 3kb of another LMR or a UMR. After examining the differences between total LMRs and the LMR subset taken by Baubec et al., we found that the LMRs that were not included in the LMR subset (“removed LMRs”) were enriched for promoter proximal locations, and had higher overall methylation and hydroxymethylation than the LMR subset analyzed in Baubec et al. (Figure 4—figure supplement 1). The higher 5mC and 5hmC levels in the removed LMRs can be observed by comparing the peak heights of MeDIP and hMeDIP in *EGFP* KD cells over total LMRs and the LMR subset (Figure 4—figure supplement 1 and Figure 4) to *EGFP* KD cells at removed LMRs (Figure 4—figure supplement 1).

*In weighing strengths and weaknesses for eLife publication one has to balance the overall quality of the descriptive data presented here with the limited insights into mechanisms and function. I am on the fence as to whether this work presents a sufficiently major advance regarding processes involved in the Mbd2/3 function.*

We feel that our findings represent a significant advance in this field. Not only do these data and analyses clarify a major error in the published literature, they also reveal novel findings regarding the interdependence between Mbd2, Mbd3, and the DNA modification machinery. Although previous studies in human cells have shown some overlap in Mbd2 and Mbd3 binding at specific genes, the interdependence between these factors has remained unexplored until now. Along with finding that Mbd2 and Mbd3 are necessary for normal chromatin localization by the other, we also found that both factors are necessary for normal levels of DNA methylation and hydroxymethylation – a very surprising finding in our opinion, and a finding that will have to be considered when mutants/KDs of MBD proteins are used in future studies. Therefore, although the first half of our manuscript is devoted to correcting an error in the literature, we feel that the novel insights into the interdependence between these marks and their “readers”, presented in the latter half of the manuscript, represent important new findings.

*By far the weakest arguments of this paper are presented in Figure 6. Similarity of all expression profiles shown in Figure 6 is not self-evident. It looks like there are some downregulated genes in the top half of the heatmap and upregulated genes in the bottom half, but it's hard to see at the used resolution and color scheme. A more formal analysis of similarity would strengthen the case the authors make. In addition, a change in the heatmap color scheme might help underscore consistency of up- and down-regulated gene sets.*

As suggested, we quantitatively assessed the alterations in gene expression in these KDs and brightened the colors in the heatmap to aid visibility. Our analyses of these data now include quantification of the degree to which the gene expression changes in each KD correlate, and we now include data from two unrelated KDs as outgroups for comparison (see response to the first point of reviewer #1 for rationale). These analyses are shown in Figure 6—figure supplement 1, and reveal significantly correlated changes among *Dnmt1, Tet1, Mbd3, Mbd2*, and *Chd4* KDs.

Regarding the colors of the heatmap, we wanted to maintain a color scheme that is visible to individuals with red-green colorblindness, and have found the yellow-blue scheme is among the best in this regard, but we brightened the blue and yellow colors to increase visibility.

*The "regulatory loop" scheme (Figure 6) as formulated in Discussion is strongly overstated. In what sense "the outcome is the same" for disruption of any component of the proposed loop when the phenotypes of Mbd2, Mbd3, and Tet KO ESCs and mice are different? Accordingly, the statement (in the Abstract) that the authors' findings "describe a regulatory loop" is too strong.*

In the revised manuscript, we toned down this statement, altering the last sentence in the Abstract along with the description of the model in the Discussion. In particular, we now discuss the functional overlap of these factors specifically in ES cell gene regulation, and acknowledge that this functional overlap is likely to change during differentiation and development (when 5hmC levels are reduced), which we do not address in this study. In addition, we have expanded our discussion regarding the phenotypes of *Mbd2* and *Mbd3* KO mice (please also see our response to the last comment from reviewer #3 and the revised text in the last two paragraphs of the Discussion). Unfortunately, there is some controversy regarding *Tet* phenotypes in the field, and since our data do not directly address this controversy, we discuss these issues only briefly in the Introduction. To clarify, phenotypic alterations upon KD or KO of *Tet* genes have been described in several studies. In two studies, KD of *Tet1* impaired ES cell self-renewal, in part because of *Nanog* down-regulation (Ito et al., 2010, Freudenberg et al., 2012). However, separate studies showed no effect of *Tet1* KO on self-renewal (Dawlaty et al., 2011; Koh et al., 2011; Ficz et al., 2011). On the other hand, *Tet1* single KO and *Tet1 Tet2* double KO ESCs are reported to exhibit a skewed differentiation profile (Dawlaty et al., 2011, Dawlaty et al., 2013), although *Tet1 Tet2* double KO mice develop to term and have only modest phenotypes (Dawlaty et al., 2013). Because of these somewhat confusing results and the fact that our ES cell data do not address any of these differentiation or developmental phenotypes, we have confined our conclusions in the revised manuscript to the roles of these factors in gene regulation in ES cells.

*Reviewer #3: I found this paper interesting, but have a few points that are mainly related to helping to resolve the controversies in this area.*

*1) The interdependence of 5mC and 5hmC levels relative to Mbd2 and Mbd3 was demonstrated with MeDIP and hMeDIP experiments (Figure 4). Since these methods depend on antibody specificity, it would be useful to support these findings with another method such as TLC on overall levels of 5mC and 5hmC.*

As suggested, we performed TLC (Figure 4) to analyze total levels and 5mC and 5hmC without the use of an antibody, as well as digests of genomic DNA with methylation sensitive or insensitive restriction enzymes (Figure 4) to look at global cytosine methylation/hydroxymethylation levels. These new data, along with dot blot assays examining the total levels of 5mC and 5hmC (Figure 4), are included in Figure 4 of the revised manuscript. The data from the antibody independent assays are consistent with the MeDIP-seq, hMeDIP-seq, and dot blots, all of which show a decrease in bulk 5mC and 5hmC levels upon depletion of Mbd2 or Mbd3.

*2) The interdependence of Mbd2 and Mbd3 binding was shown by ChIP with the antibodies that recognize endogenous proteins (Figure 5). Since Mbd2 and Mbd3 have 80% sequence similarity, it would be nice to see the genome-wide effect of Mbd2 and Mbd3 depletion on Mbd3-Flag and Mbd2-Flag occupancy with flag-tag ChIP. This would rule out cross-specificity artifacts.*

In the revised manuscript, we now include ChIP-seq data for Mbd3abc-3XFLAG and Mbd2- 3XFLAG in cells depleted of either Mbd3 or Mbd2, as requested (Figure 2—figure supplement 4 in revised manuscript). These data are consistent with our ChIP-qPCR data for Mbd3/Mbd3abc- 3XFLAG or Mbd2/Mbd2-3XFLAG in cells depleted of *Mbd3* or *Mbd2*, respectively (Figure 2—figure supplement 2; Figure 2—figure supplement 3). Together, these data confirm the interdependence of Mbd2 and Mbd3 as well as the specificity of the antibodies against the endogenous proteins.

*3) I think the paper would read better if the authors toned down the prose when comparing the Baubec et al. paper to their previous paper (Yildirim et al.) The point they are making is clear, but perhaps too clear as it has a bit of a sledgehammer aspect to it.*

As suggested, we have toned down the manuscript to more delicately explain the differences between our data and those of Baubec et al., as well as the possible reasons underlying these differences.

*4) Figure 5 shows a strong interdependency of Mbd2 and 3. How do the authors explain the phenotypic differences between Mbd3 and Mbd2 null mice?*

While chromatin binding and gene regulation by Mbd2 and Mbd3 is largely interdependent in ES cells, this interdependence is not absolute – some Mbd2 and Mbd3 binding remains upon depletion of the other (Figure 5), and their gene expression profiles do not overlap 100% (Figure 6). In addition, ES cells are a transient cell type and we do not know whether the interdependency observed in ES cells is found in various differentiated cell types. Because the functional overlap is not absolute and may not extend to all cell types, this may explain their different phenotypes during development. We have added more discussion to the manuscript regarding the differences between these phenotypes, and potential explanations for these differences (Discussion, last two paragraphs).